The following is an agreement between Journal of Systems Research (the Journal) and the submitter (the Author), governing the work currently being submitted, including the primary contribution as well as any supporting materials such as an abstract, data sets, media files, figures, or tables created by the Author and any co-authors (the Submission).

The Journal is an open access journal which means that all content is freely available without charge to readers or their institutions. Users are allowed to read, download, copy, distribute, print, search, or link to the full texts of the articles, or use them for any other lawful purpose, without asking prior permission from the publisher or the author.

**1. As consideration for publication in the Journal, the Author grants the Journal the following rights:**
1.1. A non-exclusive, irrevocable, royalty-free right to publish, reproduce, publicly display, publicly perform and distribute the Work in perpetuity throughout the world in all means of expression by any method or media now known or hereafter developed; and
1.2. A non-exclusive, irrevocable, royalty-free right to license others, including databases or printing vendors, to do any or all of the above on a non-exclusive basis.

**2. The Author warrants that:**
2.1. The Author is the author of the Submission, or is authorized to act on behalf of the author(s) and copyright holder (if different from the author(s)), and has the power to convey the rights granted in this agreement.
2.2. If the Submission has multiple authors, the other authors are identified in the Submission, and the Author will inform the other authors of the terms of this agreement.
2.3. Any textual, graphic or multimedia material included in the Submission that is the intellectual property or work of another is identified and cited in the Submission.
2.4. If the Submission reproduces any material that is the intellectual property of another, the Author has received permission to publish that material in the Submission, or the material is being incorporated based on an informed, reasonable, and good faith application of fair use.
2.5. The Submission is the original work of the Author(s). To the best of the Author's knowledge, it does not contain matter that is obscene, libelous, or defamatory; it does not knowingly violate another's right of privacy, right of publicity, or other legal right; does not contain false or misleading statements; and is otherwise not unlawful.
2.6. The Submission has not been previously published, and is not pending review elsewhere. If this is not the case, the Author will provide the Journal with information about the other locations where the Submission appears or is pending review. Prior distribution of a Submission does not mean a Submission will not be considered for publication; the Journal is primarily concerned with other appearances in similar publications.
2.7. If the Author is a student, the Author agrees to share their work and waive any privacy rights granted by FERPA or any other law, policy or regulation, with respect to the Submission, for the purpose of publication. If the Author has any student co-authors, the Author will obtain a signed copy of this agreement from those co-authors.

2.8. The Submission complies with all relevant Journal policies and submission guidelines provided on the Journal's website at the time of submission, including any policies on conflict of interest, informed consent, human and animal rights, or appropriate content.

## 3. Indemnification

The Author will indemnify and hold the Journal harmless against loss, damages, expenses, awards, and judgments arising from breach of any of the above warranties.

## 4. Author's Rights and Obligations

4.1. Nothing in this agreement constitutes a transfer of the copyright by the Author. As such, the Author retains all rights not expressly granted herein, including but not limited to, the right:

      4.1.1. To reproduce and distribute the Submission, and to authorize others to reproduce and distribute the Submission, in any format;

      4.1.2. To post the Submission in an institutional repository or the Author's personal or departmental web page.

      4.1.3. To include the Submission, in whole or in part, in another work.

4.2. If the Author distributes the Submission on another website or in another publication (as described above), the Journal will be cited as the source of first publication.

## 5. Rights for Readers

The Journal and the Author agree that the Submission will be distributed under a Creative Commons Attribution-NonCommercial 4.0 International License (CC BY-NC 4.0), or other later version of the same license, that allows others to copy, distribute, translate, adapt, and build upon the Submission, as long as they provide appropriate credit to the author(s) and do not use the Submission for commercial purposes. Anyone who uses or redistributes the Submission under this license must indicate any changes that were made, must link to the license, and cannot imply that the author(s) endorse them or their use. More information about this license is available at https://creativecommons.org/licenses/by-nc/4.0/.

## 6. Termination

The Author agrees to the terms of this agreement for the Submission being considered for publication. If the Submission is declined, this agreement is terminated.

| Paper Title: | SoK: The Great GAN Bake Off, An Extensive Systematic Evaluation of Generative Adversarial Network Architectures for Time Series Synthesis |
| --- | --- |
| Name of the Author:
Date: | Mark Leznik |
| | 23.09.2022 |

| Signature: | |
| --- | --- |

# SoK: The Great GAN Bake Off, An Extensive Systematic Evaluation of Generative Adversarial Network Architectures for Time Series Synthesis

MARK LEZNIK
*Ulm University*
mark.leznik@uni-ulm.de

ARNE LOCHNER
*Ulm University*
arne.lochner@uni-ulm.de

STEFAN WESNER
*University of Cologne*
wesner@uni-koeln.de

JÖRG DOMASCHKA
*Ulm University*
joerg.domaschka@uni-ulm.de

## Abstract

There is no standard approach to compare the success of different neural network architectures utilized for time series synthesis. This hinders the evaluation and decision process, as to which architecture should be leveraged for an unknown data set. We propose a combination of metrics, which empirically evaluate the performance of neural network architectures trained for time series synthesis. With these measurements we are able to account for temporal correlations, spatial correlations and mode collapse issues within the generated time series.

We further investigate the interaction of different generator and discriminator architectures between each other. The considered architectures include recurrent neural networks, temporal convolutional networks and transformer-based networks. So far, the application of transformer-based models is limited for time series synthesis. Hence, we propose a new transformer-based architecture, which is able to synthesise time series. We evaluate the proposed architectures and their combinations in over 500 experiments, amounting to over 2500 computing hours. We provide results for four data sets, one univariate and three multivariate. The data sets vary with regard to length, as well as patterns in temporal and spatial correlations.

We use our metrics to compare the performance of generative adversarial network architectures for time series synthesis. To verify our findings we utilize quantitative and qualitative evaluations. Our results indicate that temporal convolutional networks currently outperform recurrent neural network and transformer based approaches with regard to fidelity and flexibility of the generated time series data. Temporal convolutional network architecture are the most stable architecture for a mode collapse prone data set. The performance of the transformer models strongly depends on the data set characteristics, it struggled to synthesise data sets with high temporal and spatial correlations. Discriminators with recurrent network architectures suffer from vanishing gradients. We also show, that the performance of the generative adversarial networks depends more on the discriminator rather than the generator.

## 1 Introduction

Machine Learning (ML) is a wide research field nowadays, offering new possibilities and promising solutions for a range of data driven problems. One of the main limitations in ML is the availability and accessibility of large problem specific data sets, which hinders the performance and generalizability of the trained models. Using *generative models* for data generation and augmentation is a growing field of research aiming to counteract this issue. Specifically in the image domain, it has been shown to improve the accuracy of the models using the synthetic data [34]. The possibility of generating or extending data sets mitigates the problems of data acquisition or limited data sets due to security and privacy concerns, or restrictions (e.g., GDPR) [10, 27]. 60% of the data used for the development of ML driven applications are predicted to stem from synthetic data generation by 2024[1]. *Synthetic data* can be used to extend unbalanced data sets, or to augment failure cases and anomalies. In order to utilize the synthesised data, synthetic data generation must fulfill strict properties. Generated samples should be drawn from the same data distribution as the underlying ground truth data [27]. Further, a non-biased heterogeneous data set must be the result of any synthetic data generation. In cases where privacy concern is the main limitation, synthetic data generation must not expose critical information of the real data. Synthesised data sets are relevant for different data types such as audio, text, images and also time series. Synthetic time series, which are the focus of our work, aid research by being used to pre-train models, obfuscate sensitive information or augment unbalanced data sets. Synthetic data applications are widespread in several domains, for example, generated time series allow system administrators to modify their resource allocation in order to improve the overall performance of their provided services [30].

Currently, a *Generative Adversarial Network (GAN)* [11] is the predominant architecture used for data synthesis. The main issue within the research landscape of time series generation, however, is an the lack of comparability and evaluation between the proposed architectures (it is unfortunately not exclusive to it). It is unclear, which neural network architecture is the most suitable for an unknown data set. Hence, in this work we focus on evaluating the state of art in time series synthesis GAN architectures.

Time series analysis can be challenging, since the data contains spatial and temporal correlations, often not clearly captured in visualisations. Spatial correlation pattern is present, if

---

[1] Andrew White, Gartner

multiple channels of the same time series influence each other. A temporal correlation refers to observations at a present time point correlating with observations in the past.

With this in mind, our main goal in this work is to answer the following question:

- What are the most suitable GAN architectures to capture spatial and temporal correlations of time series?

- How can the architectures and the generated data be compared and evaluated against each other?

As part of this work, we develop an approach to make the performance of multiple GAN models comparable. We evaluate nine different GAN architectures, in over 500 experiments amounting to a total of over 2500 computation hours. We use measurements from the area of time series analysis and time series similarity to evaluate the success of the time series synthesis. Our main contributions lie in the insights as to which GAN architecture fits which task best, as well as a comprehensive pipeline to evaluate synthetic time series fidelity.

The remainder of this work is organized as follows. Section 2 provides the fundamentals of time series evaluation, neural networks and data synthesis. Section 3 gives an overview of the state-of-the-art for synthesising and evaluating time series data, as well as neural network comparison work. Section 4 describes the methodology, followed by the data used in our work in Section 5. We then detail our approach for synthesizing and comparing time series data in Section 7. The results of this work are presented in Section 7 and are discussed in Section 8. Section 9 concludes and summarizes our work.

## 2 Background

### 2.1 Data Synthesis

In order to evaluate and analyze data generated by ML applications, knowledge about its probability distribution is required. Generating desired artificial data with certain statistical characteristics by estimating the underlying distribution function has been a focus in research for over a decade [11]. Probability distribution estimation is relevant in different domains and for different data types such as audio, text, images and time series. When knowledge about the underlying distribution function is present a deep generative model such as the deep Boltzmann machine [32] can be utilized. However this requirement cannot always be satisfied [11]. Other approaches such as Noise-Contrastive Estimation (NCE) and Variational Autoencoders (VAEs) have been proposed for data synthesis task, but both have limitations while estimating probability distributions [11]. While VAEs have been successfully used for end tasks such as anomaly detection [37], the research field has not experienced a push in recent years, similar to

that provided by image synthesis for GANs. Hence, in this work, we focus our attention on GANs, as they are, as of now, the go-to architecture approach for data generation tasks. In the following, we provide further background of GANs as well as possible modifications and optimization approaches.

#### 2.1.1 Generative Adversarial Networks

In 2014, Goodfellow et al. [11] suggested GANs to implicitly estimate the probability distribution of training data via an adversarial process. A GAN framework consists of two neural networks, which compete against each other: a *generator*, which generates synthetic data and a *discriminator*, which aims to distinguish between real data and the generated data of the generator. The generator is trained to transform a fixed input noise distribution into the underlying ground truth data distribution. A uniform or normal distribution is commonly chosen for this noise distribution $p_z(z)$, also called latent space. The process of generating data by mapping a noise vector sampled from the latent space is noted by $G(z; \theta_G)$ with $z \sim p_z(z)$. Considering the present data distribution $x \sim p_{data}(x)$ and $z \sim p_z(z)$, $D(x, \theta_D)$ and $D(G(z; \theta_G), \theta_D)$ represent the differentiation process of the discriminator. A value function is optimized by the GAN during the training process, in order to minimize the Kullback-Leibler divergence between the underlying distribution and the estimated distribution. The Kullback-Leibler divergence measures the dissimilarity between two probability distributions. It achieves its minimum value, when the underlying and estimated distribution are exactly similar. This value function is displayed in Equation 1. $V(D, G)$ contains two parts for its calculation. The first part considers how accurate the discriminator classifies real data as "non-fake". For the second part, the discriminator is asked to classify data which is synthesised by the generator. The task of the generator to synthesise data, which the discriminator cannot distinguish from real samples, is equivalent to maximizing $D(G(z))$. The generator cannot influence the first part of the equation directly. This leads to training the generator to minimize Equation 2. As the discriminator itself can influence both parts of the equation $V(D, G)$ it aims to maximize Equation 3.

$$V(D, G) = \mathbb{E}_{x \sim p_{data}(x)}[\log D(x)] \\ + \mathbb{E}_{z \sim p_z(z)}[\log(1 - D(G(z)))] \quad (1)$$

$$\min_G V(D, G) = \mathbb{E}_{z \sim p_z(z)}[\log(1 - D(G(z)))] \quad (2)$$

$$\max_D V(D, G) = \mathbb{E}_{x \sim p_{data}(x)}[\log D(x)] \\ + \mathbb{E}_{z \sim p_z(z)}[\log(1 - D(G(z)))] \quad (3)$$

The optimization of these neural networks is realized by applying gradient decent and ascent utilizing back-propagation with respect to trainable parameters ($\theta_G$ and $\theta_D$). The training procedure consists of alternately training the generator and discriminator in a minimax game. Goodfellow et al. [11] state, that with this value function and optimizing approach, the generator is able to recover the data distribution, as long as enough training iterations are provided.

The architecture suggested by Goodfellow et al. [11] is shown in Figure 1.

The resulting trained GAN should be able to synthesise data sets with two properties [27]:

1. *High fidelity:* The generated samples should be drawn from the ground truth data distribution.

2. *High flexibility:* The generated data set should consist unique data samples. This can only be achieved, if no mode collapse occurs while training the GAN.

### 2.1.2   Generative Adversarial Modifications

The original GAN architecture as proposed by Goodfellow et al. [11] is prone to training instabilities. Specifically, non-convergence, mode collapse and vanishing gradients limit the performance of the GAN architecture [1, 12, 29]. Non-convergences occurs, when the GAN model parameters oscillate and never converge. In contrast, vanishing gradients can occur, when the discriminator does not provide sufficient feedback to the generator, which hinders the learning process. If the generator is only able to generate samples with low variety, a mode collapse is present. To counteract these problems several modifications to the value function [1, 12], the adversarial training and neural network architectures [5, 19] have been proposed.

**Loss Functions**   Goodfellow et al. [11] recognized themselves, that their initial loss function as described in Equation 1 can lead to unstable training in early iterations. For more stable training they propose, that the generator should not try to minimize $\mathbb{E}_{z\sim p_z(z)}[\log(1-D(G(z)))]$ but maximize $\mathbb{E}_{z\sim p_z(z)}[\log(D(G(z)))]$ instead.

One limitation of the GAN training proposed by Goodfellow et al. [11] is that a good balance between the training of the discriminator and generator is required. With the original GAN setup, the discriminator can not be trained until optimum, because the generator would not receive sufficient gradients in this case. To counteract this issue Arjovsky, Chintala, and Bottou [1] suggest to use a Wasserstein GAN (WGAN). The WGAN utilizes a critic instead of a discriminator. The critic has a linear output function instead of classifying the generated and real samples into a range of $[0;1]$. The critic is trained to classify the generated data with small values and the training data with high values. To train the critic, the authors show that it is required to keep the weights of the

Artificial Neural Network (ANN) in a compact space. To achieve this constraint, they propose to clip the weights of the ANN. Other approaches include using gradient penalty terms to enforce this constraint [12]. This Wasserstein loss function allows to train the discriminator to optimum, which still provides sufficient gradients to the generator. In general, this WGAN leads to more stable GAN training.

### 2.2   Time Series Synthesis

In order to discuss time series synthesis, a brief definition of time series itself is necessary. A time series is an ordered sequence of data points [7]. These data points represent observations of specific values at given timestamps. In our work, we assume time series with equidistant data points. Such discrete-time data can be denoted as $X = \{x_1, x_2, \ldots, x_n\}$ with $n$ indicating the length of the time series. If $x_i \in \mathbb{R}$, the time series consists of only a single value at each time-step and is called univariate. However, it is common that $x_i \in \mathbb{R}^d$, where each time-step contains multiple channels. Such time series are called multivariate time series [13], which are the main focus of this work.

While the concept of GANs is not limited to any domain, the computer vision research field was a pioneer utilizing GANs for image synthesis tasks [5, 11, 19]. As mentioned by Goodfellow et al. [11] a straightforward implementation of a GAN would consist of two Multi Layer Perceptron (MLP) networks for the generator and discriminator part. This architecture was substituted by Convolutional Neural Networks (CNNs), which have multiple benefits for computer vision tasks compared to MLP networks [5, 19, 31].

Lin et al. [27] note that GAN architectures, that were designed for computer vision tasks are not able to be used effectively for time series synthesis tasks. The authors further argue that CNNs cannot detect the complex correlations of time series accurately. This results in a low fidelity of the generated samples [27]. Generally speaking, time series have the following properties, that have to be considered when designing a GAN architecture for time series synthesis: Temporal Correlations (e.g., diurnal cycles and Long Range Dependencies); and Spatial Correlations.

## 3   Related Work

This section provides an overview of approaches to synthesise time series data. We discuss ANN-based approaches and their respective architectures. For this, we focus on Recurrent Neural Network (RNN) [10], Temporal Convolutional Network (TCN) [36] and transformer [26] architectures to synthesise time series data, discussing their limitations and applications. We then cover approaches to compare the performance of different ANN models, in general, followed by time

---

[2]Icons designed by www.freepik.com

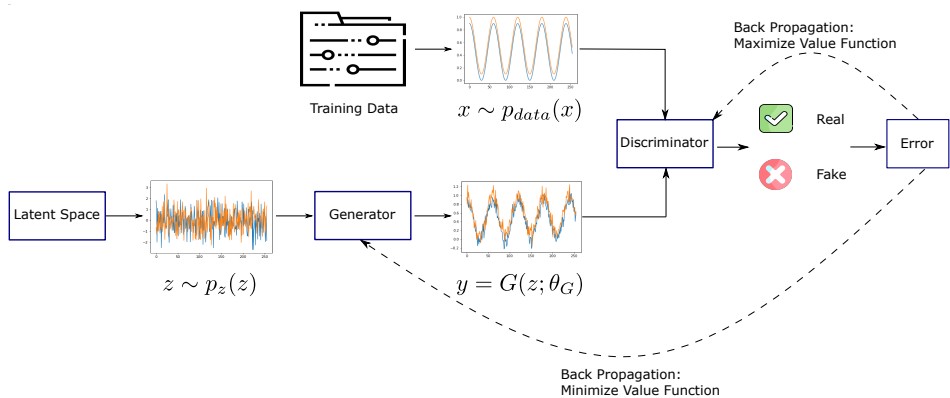

Figure 1: GAN architecture suggested by Goodfellow et al. [11] used for data distribution estimation. The GAN models consists of a generator and discriminator, which play a minimax game to optimize the model.[2]

series synthesis evaluation. These methods can be categorized into time series analysis, descriptive statistics, distribution comparison, supervised evaluation and downstream tasks. Figure 2 displays the different fields utilized in the current state of the art. Noticeably, there exists no common evaluation method in the context of time series synthesis, which limits the comparability of results.

### 3.1 Time Series Synthesis using Generative Adversarial Networks

The introduction of *GANs* [11] led to major improvements for synthetic image generation in recent years. In the field of computer vision, GANs can be trained to generate photo-realistic images [19, 20]. Currently, *CNNs* are utilized by most GAN architectures [19, 20].

Based off of that, GAN approaches for time series synthesis [6, 10, 27, 36] have been successfully proposed. However, time series data holds different characteristics when compared to images [27]. CNN-based GAN architectures were not able to be used for time series tasks without any modifications [27]. The receptive field size of CNNs increases linearly compared to the number of layers used [3]. Based on the required receptive field size, synthesising time series data with long temporal correlations is limited. Additionally, when synthesising multivariate time series, the GAN architecture has to capture correlations between the different time-dependent variables of the time series as well.

**Recurrent Neural Network based Approaches**   Esteban, Hyland, and Rätsch [10] proposed to utilize the potential of *RNNs* to modify the GAN architecture for time series tasks. The authors implemented the RNN by using Long Short-Term Memory (LSTM) layers [15]. With this approach Esteban, Hyland, and Rätsch [10] argue, that the generator and discrim-

inator part of the GAN are able to process time series more effectively by accounting for temporal correlations.

The authors utilized this architecture to synthesise sine waves, the MNIST data set and patient information of an intensive care unit, which was provided by the Electronic Intensive Care Unit (eICU) Collaborative Research Database. The authors synthesised patient time series, which contains regularly-sampled data measured by bedside monitors. The MNIST[4] data set contains grey-scale images of hand-written digits with a image size of $28^2$. The authors flattened the MNIST images into a vector of size 784 and interpreted it as a time series. The used data sets vary with regard to spatial and temporal correlations.

To evaluate the success of the time series synthesis, Esteban, Hyland, and Rätsch [10] utilized a downstream task, methods for distribution comparison and a latent space analysis. A downstream task describes an approach to use the generated samples of a GAN to either train or evaluate a separate machine learning algorithm. It is then trained or evaluated on the real data set. If the generated samples share similar characteristics to the real data set, the performance of the downstream task algorithm should not vary.

Additionally, the authors evaluated the fidelity of the generated data by calculating the maximum mean discrepancy. The maximum mean discrepancy quantifies if two sets of time series were generated by the same distribution. For this task, a kernel function is required, and the authors used the Euclidean Distance (ED) to compute the similarity score. While the maximum mean discrepancy quantifies how similar the generated examples are on average to the training data set, this can lead to issues if the underlying data distribution is heterogeneous with multiple clusters. In this case it might be more suitable to only consider the similarity to the closest neighbours. Additionally, the authors did not evaluate how similar their generated data are to the training data with regards to irregularities, noise and seasonalities.

---

[3]Icons designed by www.freepik.com

[4]http://yann.lecun.com/exdb/mnist/

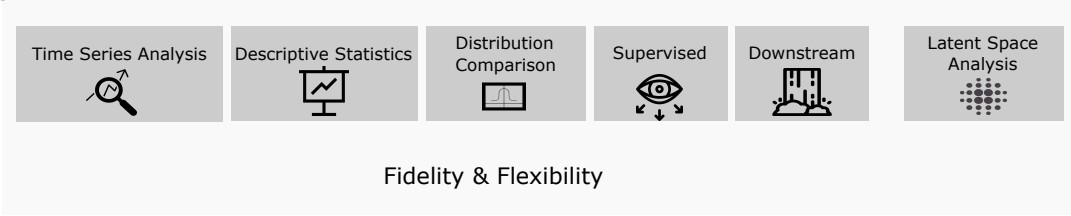

Figure 2: Overview of evaluation approaches for time series synthesis. Overall, authors used methods from the field of time series analysis, descriptive statistics, distribution comparison, downstream tasks and latent space analysis to evaluate the performance of their neural networks.[3]

Lin et al. [27] extended the architecture proposed by Esteban, Hyland, and Rätsch [10] and introduced DoppelGANger. DoppelGANger was trained to synthesise data sets with temporal and spatial correlations as well as univariate and multivariate time series. The authors synthesised the univariate Wikipedia Web Traffic (WWT) data set, which tracks the number of daily views for various articles. The proposed architecture counteracts the limitations of RNNs, which fail for long sequences due to exploding/vanishing gradients.

DoppelGANger generated multiple time series points from a single output of the RNN by using an MLP. Thereby, DoppelGANger can process longer and variable sequences more effectively. Lin et al. [27] achieved this by extending the GAN output by a flag, which indicates if the time series is complete. While Esteban, Hyland, and Rätsch [10] coupled the generation of attributes and features, Lin et al. [27] noticed, that it is beneficial to isolate the generation process of attributes and features. With this architecture, it is possible to later retrain the attribute MLP to hide certain characteristics of the training data. This can make membership inference attacks, which try to determine if a record was part of the training data for an ML model, more difficult. Furthermore, DoppelGANger utilizes an auxiliary discriminator, which only classifies the attributes of input samples. Thereby at early steps the generator can focus on learning realistic attributes. This improves the unstable training of the generator. Lin et al. [27] remark, that data with high variance was more prone to mode collapse. Therefore they normalized the time series in order to achieve more stable training. A min/max generator is utilized to generate non-normalized time series data. To evaluate the success of the time series synthesis, Lin et al. [27] compared the generated data with the underlying data set with regard to temporal correlations, spatial correlations and a downstream task. They compared their results to the benchmarks of other baseline models. Those baseline models included Markov-models, non-linear auto regressive model, RNN in a non-GAN context and an MLP-based GAN. The authors did not compare their benchmarks to other RNN-based GAN models. To account for temporal correlations, the authors compared the auto-correlation of generated examples and training examples.

While RNNs improved the performance of GANs for time series generation, they still come with downsides with regard to stable gradients, memory consumption and training time.

**Temporal Convolution Network based Approaches**  To counteract limitations of RNNs, Bai, Kolter, and Koltun [3] analyzed the possibility of modifying CNNs for time series tasks. Bai, Kolter, and Koltun [3] proposed *TCNs*. With an introduction of a dilation factor at each layer, it is possible to achieve a flexible receptive field size. Additionally TCNs greatly reduce computation time when compared to RNNs. Bai, Kolter, and Koltun [3] provide empirical results that TCNs can outperform RNNs for time series tasks. However the authors did not provide empirical results for data synthesis tasks. Wiese et al. [36] introduced QuantGAN, a deep generation network used for generation of financial time series. The generator and discriminator part contain TCN layers to capture long range dependencies, such as the presence of volatility clusters [36]. QuantGAN is trained to generate univariate time series data, which represent the log value of the percentage change of a share. Wiese et al. [36] argue, that the training of a TCN based GAN is more stable with regard to vanishing/exploding gradients than a RNN based approach. For evaluation purposes the authors compared the distributions by calculating an earth mover distance, which can be seen as a similar measurement as the maximum mean discrepancy used by Esteban, Hyland, and Rätsch [10]. To account for temporal correlations in the time series data the auto correlation and a leverage effect score is computed. The authors did not compare their benchmarks to RNN-based GAN approaches.

**Transformer based Approaches**  The usage of transformer models for time series tasks is limited so far. Additional temporal processing is required to capture time-varying relationships for time series Sequence-to-Sequence (Seq2Seq) tasks.

Lim et al. [26] modified the transformer architecture for multi-horizon time series forecasting. The transformer based architecture was trained to predict multiple time steps in ad-

---

[4]Icons designed by www.freepik.com

vance. The model considered static information, observed information and known future information as inputs. Lim et al. [26] used static covariate encoders that allow the network to condition temporal forecasts on static metadata. The resulting Temporal Fusion Transformer (TFT) utilized LSTM encoder and decoder components to capture time-varying relationships before applying a temporal self-attention component. The authors argue, that the LSTM component is used for local processing, but they do not discuss, if this architecture should be seen as an approach to counteract vanishing gradients issues of RNN or a modification to the transformer model for time series tasks. Gating components were implemented to allow the network to skip parts which are unnecessary for a given data set.

However, it is still unknown how transformer architectures can be utilized for data generation tasks via GANs for time series. Recently Jiang, Chang, and Wang [18] implemented a GAN architecture called TransGAN to generate images, which is based on a transformer generator and discriminator. As previous research suggests, image GAN architectures do require modifications for time series tasks [27].

## 3.2 GAN Performance Comparison

In the computer vision domain, Lucic et al. [28] investigated the performance of different loss functions for the task of image synthesis. The authors utilized four common computer vision data sets for this task. As the focus was on evaluating different loss functions, the architecture of the GAN models was fixed. The authors performed a parameter search to find the best configuration for each model. This included optimizer specific parameters, loss function specific values and the number of times the discriminator is trained before updating the generator. The authors utilized a random search and computed the Fréchet Inception Distance (FID) between generated and training examples every five epochs. After finding the best configuration for each model, the training of these selected models is re-run to estimate the stability of the GAN models.

With the introduction of TCN networks, Bai, Kolter, and Koltun [3] provided empirical results for their performance. They compared the TCNs to multiple RNN networks, including LSTM-based ANNs. The authors evaluated the performance of TCN models with minimal tuning for multiple sequence modeling tasks, commonly used for benchmarking purposes. The authors did not provide empirical results for data generation by GANs. Bai, Kolter, and Koltun [3] performed a grid search for the parameters of the RNN-based models. The parameters search included optimizer parameters and a limited amount of architectural parameters. For the TCN models no parameters search was performed and the kernel size and number of layers was only modified for each task to adapt the required receptive field size.

## 3.3 Data Synthesis Evaluation

Accurate evaluation metrics are required to judge the performance of synthesised data. This necessity is amplified by the unstable training process of a GAN. For image generation, common evaluation metrics such as the Inception Score (IS) [33] and FID are used to compare the performance of different GAN architectures. The IS considers the variety and quality of the generated images to measure how realistic the learned distribution is. For this task a pre-trained Inception classifier network is utilized. The FID can detect mode collapse issues in image generation and is a suitable metric to account for image diversity. However, Barratt and Sharma [4] arguem that the IS is an undesirable metric to evaluate GANs in the computer vision field. One benefit when working with GANs in the computer vision field is that, the performance of GANs can be roughly estimated by qualitative evaluation. This pattern can be seen by most researchers simply providing example images in their work. However, for time series generation, such qualitative measurements are limited and quantitative measurements such as the IS and FID have not yet been introduced.

For time series data, it is common to evaluate if temporal correlations were learned by the GAN. For this purpose Lin et al. [27] and Wiese et al. [36] compared the auto correlation of generated data to the underlying training set. Esteban, Hyland, and Rätsch [10] did not provide any evaluation that aims to measure if temporal correlation were learned. Lin et al. [27] used the length distribution to measure the quality of the generated data. This measurement is only suitable for GAN models, which can generate time series with various lengths. To account for spatial correlations, Leznik et al. [24] also compared the correlation coefficient of the generated data. The authors synthesised a Content Delivery Network (CDN) data set from a production environment, which contains time series with high temporal and spatial correlations. Comparing descriptive measurements of generated time series as can also provide insights [24]. Leznik et al. [24] utilized different entropy measurements to compare the generated time series with regard to information and noise. Using downstream tasks can also provide insight if the GAN was able to learn the underlying data distribution correctly. To avoid privacy concerns the GAN model should not overfit the training data [10, 27]. Therefore, an analysis of the latent space via interpolation can be used [10]. For this purpose the difference to the nearest neighbours can be analyzed [27].

Overall it is noticeable, that the presented work used different measurements to evaluate the success of the time series synthesis. Not a single metric was consistently used in all listed approaches. Additionally, no paper compared their approach to another approach utilizing their suggested metrics. GAN specific problems, such as mode collapse, were neither investigated, nor quantified. Some authors provided benchmarks for baseline models, however no results were provided

by comparing different GAN architectures. These limitations are partly caused by the authors synthesising different data sets. With these limitations, it is unknown, which neural network architecture is the most suitable for an unknown data set.

## 4 Methodology

As seen in Section 3, there is no common approach to evaluate and compare the performance of different GAN models. Therefore, we propose an approach to use a combination of time series analysis and distance measurements to empirically evaluate the performance of the data synthesis task. With these measurements, we can compare generated time series with regard to fidelity and flexibility. Generated samples of the GANs are utilized to compute the measurements for evaluation purposes. These computed values are then compared to the same measurements computed on a set of the training examples.

Based on the current state of the art, we consider architectures based on RNN, TCN and transformers. These are the go-to architectures for time series tasks. We investigate the interaction of GAN models with mixed network architectures based on the mentioned building blocks.

To improve the generalizabilty of the results, we perform a network parameter search for each architecture before comparing its performance. This network parameter search includes architecture specific parameters, e.g., number of channels in a convolutional layer. Further, GAN specific parameters which influence the training are investigated, such as the alternate training setup. Based on this parameter search, we narrow the scope and further optimize the best performing GAN models. The performance of the data synthesis is then evaluated.

**Evaluation Metrics**    We utilize multiple metrics to compare the performance of different GAN models. We select metrics to account for fidelity and flexibility of the generated time series. We lay a specific focus to evaluate similarity between the generated time series and the training data with regard to temporal correlations, spatial correlations, noise and uncertainty. In contrast to the suggested evaluation approaches listed in Section 3, we also consider metrics to account for possible mode collapse issues.

**Network Architectures**    Based on the state of the art, we consider RNN, TCN and transformer architectures for the comparison task. The RNN architectures we use are based on Esteban, Hyland, and Rätsch [10], Lin et al. [27] and Leznik et al. [24]. These authors utilize an LSTM block to process the time series and MLP layers to achieve the desired output dimensionality.

For the TCN networks, we utilize the modified convolutional layers proposed by Bai, Kolter, and Koltun [3]. Wiese

et al. [36] leveraged the TCN layers for a univariate time series synthesis task. We modify this architecture to generate multivariate time series.

The application of transformer models for time series tasks is limited so far, and does not offer viable transformer-based GAN architectures for time series synthesis. In this work, we modify the TFT architecture proposed by Lim et al. [26] for the task of time series generation.

**Parameter Search**    To improve the generalizabilty of the results we perform a parameter search for every architecture before comparing the performance. For the parameters search we consider architecture specific parameters, such as the number of layers, as well as GAN training specific parameters, such as the alternate training setup. Based on the that, we perform a grid search which evaluates all possible parameter combinations. We limit the scope of this parameters search based on parameters suggested by previous research (cf. Section 3). Based on this parameter search, the GAN models with selected parameters are trained and the success of the data synthesis between the different architectures is evaluated. Our approach is illustrated in Figure 3.

**Data Sets**    Data sets vary with regard to possible temporal correlations, for multi-variate time series data, spatial correlations are also a factor. With these information in mind, we want to evaluate the performance of the different GAN architectures based on multiple data sets, which cover different characteristics and domains. Additionally, we introduce rare events in the form of anomalies into one of the data sets, to show the ability of network to learn the underlying data distribution including anomalous events. We provide empirical results for data sets with varying temporal correlations and spatial correlations. While most time series data sets include multi-variate time series, we also consider univariate time series without any spatial correlations. For the time series data characteristics, we analyze the data sets with the help of time series decomposition and the Pearson correlation. For the initial prototyping phase, we utilize an artificially created time series data set.

## 5 Data Sets

We consider four data sets with different characteristics. We use an artificial sine data set to provide empirical results based on available ground truth information, following the approach of Esteban, Hyland, and Rätsch [10]. We also provide empirical results for an univariate data set. The remaining two multivariat data sets, differ in regard to spatial and temporal correlations as well as their respective domains. The chosen data sets was shown to be used in synthesis tasks by GAN architectures (cf. Section 3).

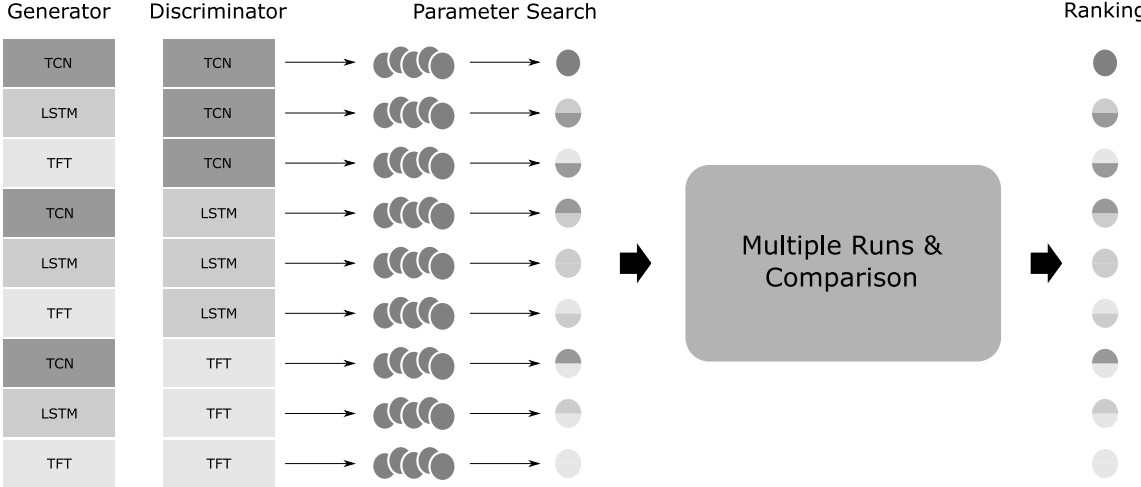

Figure 3: Process used to evaluate and compare the different architectures for the research question. First, we perform a parameter search to find the optimal configuration for each architecture. We then compare the best performing architectures.

In line with Leznik et al. [24] this work focuses on synthesising time series with a fixed length. According to Lin et al. [27], RNN-based GAN models struggle to synthesise time series with more than 500 time steps. Therefore, for all data sets, time series with a modest length of 256 data points are used to train the GAN model. This should not hinder the performance of RNN-based architectures due to vanishing gradient problems.

As suggested by Lin et al. [27] all data sets are normalized to a range of $[0; 1]$ to prevent mode collapse issues.

In the following, the data sets are described in detail. A Fast Fourier Transformation (FFT), a time series decomposition and Pearson correlation computations are utilized to calculate temporal correlations, seasonality and spatial correlations in the data.

## 5.1 Sine and Cosine Waves

For the artificial data set we use sine and cosine waves with varying frequencies. Each time series consists of a periodic wave with a single-frequency component. After normalizing the data, each time series can be well defined by its frequency component. All periodical time series contain at least one completed period and a maximum of five. In order to generate a multivariate time series, we shift the periodic wave along the y-axis by a constant value at each time point for the second channel. This creates a data set with a perfect spatial correlation.

After applying a min-max scaler on each periodic wave the amplitude of all training examples are equivalent and the samples only vary with regard to their frequency component. The amplitudes of the first channel are in range of $[0; 0.9]$ and of the second channel in range of $[0.1; 1]$. The periodical signals have a fixed length of 256, and a minimal period length

is defined before sampling training data from this artificial distribution. Another major benefit when working with sine and cosine waves is, that in the initial protoyping phase, the performance of the GANs can be judged by qualitative measurements. For the sine and cosine data set 10000 samples are drawn from an uniform distribution which defines the frequency component.

## 5.2 Content Delivery Network Cache Utilization

In line with Leznik et al. [24], we use the public accessible information about downloading content from cache to serve users of a CDN of the British Telecom (BT) from three different backbone locations to evaluate the performance of the different architectures for a data set with high spatial and temporal correlations. The information is extracted from inner-core nodes in London, UK. The cache access is measured in bits per seconds and was sampled every 1200 seconds for the time span between 2016 and 2017. This leads to a time series length of 19728. For security reasons, time stamp information is obfuscated, and the cache utilization is normalized. We detected the high temporal correlations by applying an FFT on the time series data (cf. Figure 4). Throughout 2016 to 2017, a strong daily cycle is present. The same patterns are present in each channel of the time series.

We use a sliding window approach to generate multiple time series from the original data. To generate a data set with the maximum number of examples, we shift the sliding window by a single time series step at each iteration.

Extracting time series with a length of 256 with the given resolution of a sample every 20 minutes, results in each training examples covering roughly 3.5 days. As the CDN data set is already normalized, no min-max scaling is applied.

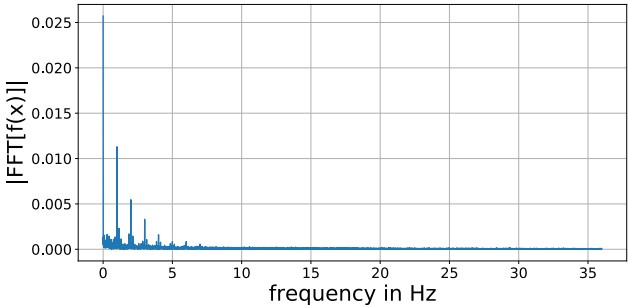

Figure 4: Results of FFT applied on a single channel of the CDN data set. The results of the FFT imply, that a strong daily seasonality is present. The X-axis represents the frequency component, with $frequency = 1$ being the daily seasonality.

Assuming the original time series is defined as $X \in [0;1]^{19728 \times 3}$, the generation of the training data set $T$ with the sliding window approach can be noted as:

$$T = \{\{x_i, x_{i+1}, \ldots, x_{i+255}\} \mid 0 \leq i \leq 19472\} \quad (4)$$

with $x_i \in [0;1]^3$ representing the observed values at time point $i$.

Examples of the extracted training data set $T$ are shown in Figure 5.

### 5.2.1 Rare Events

We manually introduced rare events in the form of anomalies into the CDN data set. Hereby, we consider point and collective anomalies, as those generally allow for a visual inspection and evaluation of the data. An individual data point that is considered anomalous with respect to the rest of the data is a point anomaly. As a real-life example, a system shutdown or a sudden spike out of the scope of the currently measured system performance is considered a point anomaly. A collection of data points that is anomalous in respect to the entire data set is known as a collective anomaly. Yet, here individual points by themselves may or may not be considered as an anomaly.

In our case, we have used sudden load spikes in the measurements as well as missing measurements caused, e.g., by a system shutdown as a ground truth.

### 5.3 Electronic Intensive Care Unit

The second multivariate multivariate data set presents less temporal correlations. Time series data of patients from multiple critical care units in 2014 and 2015 throughout the United States are used. The data is provided be the eICU Collaborative Research Database and was already used by Esteban, Hyland, and Rätsch [10]. Of the available data, the periodical measurements, which are collected from bedside sensors are

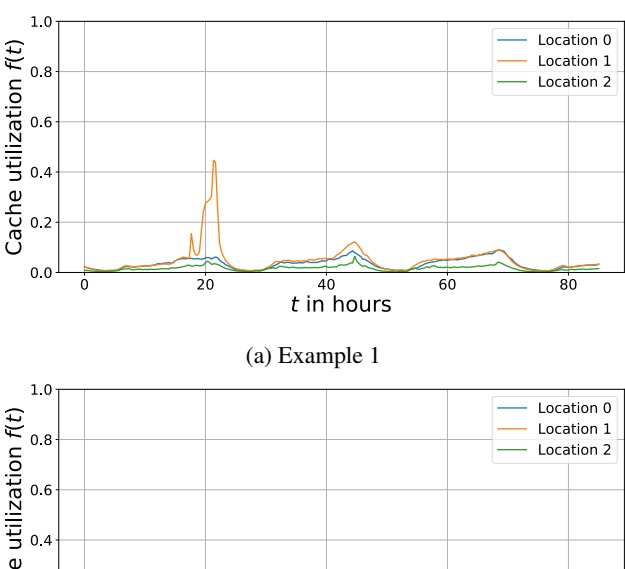

(a) Example 1

(b) Example 2

Figure 5: Visualization of two examples from the training data set $T$, which was created by applying the sliding window approach with a fixed length of 256 on the CDN data set. In each plot the multivariate time series are displayed. Each channel represents the normalized cache utilization of a single location. Each time series covers a time span of roughly 3.5 days. The diurnal cycles and high correlation between the cache utilization of different locations can be seen.

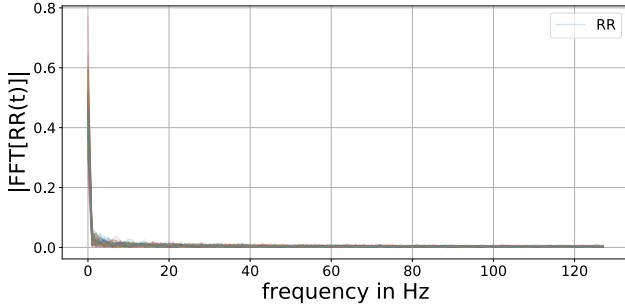

Figure 6: Result of FFT analysis applied on the respiration rate channel of 1000 examples of the eICU data set. Compared to Figure 4 no obvious patterns can be seen. The highest peak is achieved in the lowest frequency component. This frequency bin contains non-periodic components.

synthesized as part of this work. The database archived these measurements as 5 minute median values.

The GANs are trained to synthesize the heart rate, respiratory rate and the $O_2$ saturation. To detect seasonalities and cycles in the eICU data set, the time series data are transformed into the frequency domain via FFT. The result of applying an FFT on the respiration rate of 1000 patients data can be seen in Figure 6 and does not indicate any obvious seasonality in the eICU data set. The same pattern is present for the heart rate and $O_2$ saturation channels. The FFT analysis found a relevant low frequency component, which can be attributed to non-periodical components in the time series such as an overall trend increase. With time series only covering less than a day, patterns such as diurnal trends can not be detected. In general, the multivariate time series do not contain as strong cycles and seasonalities as the CDN data set.

The eICU data set can vary with regard to time series length and data quality. The GANs are trained to synthesise time series with fixed length of 256. Therefore only observations of patients, which stayed at for least 21 hours in the intensive care unit are used. As some time series can exceed the time series length of 256 we cap all remaining time series to a length of 256. To counteract anomalies in the data set, only patient data without missing values are considered. These filtering approaches resulted in 7917 relevant patient time series. The resulting training data set can then be defined as $X \in [0; 1]^{7917 \times 256 \times 3}$. Examples of this training set $X$ are displayed in Figure 7.

## 5.4 Wikipedia Web Traffic

To evaluate the architectures for a univariate context without any spatial correlations, we use the WWT data set, as previously done by Lin et al. [27]. The WWT data sets contains the daily views of different Wikipedia articles. From this data

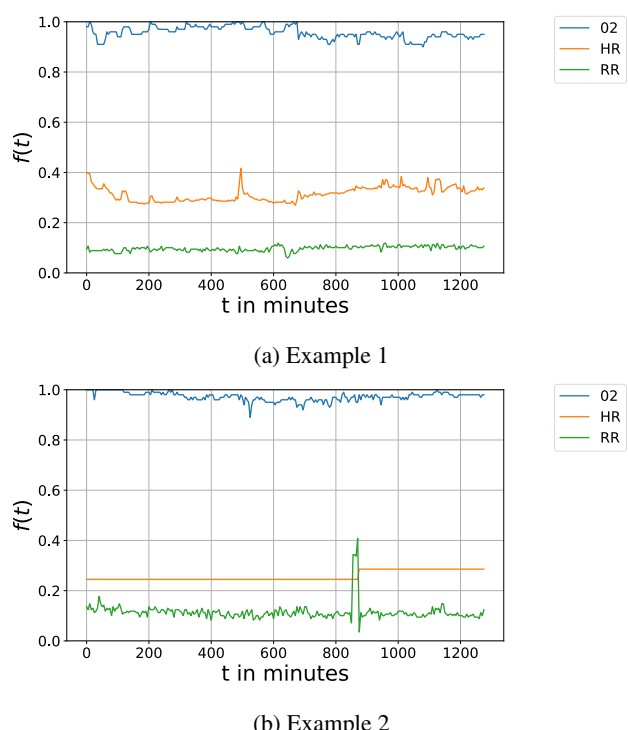

(a) Example 1

(b) Example 2

Figure 7: Visualization of two examples from the training data set $X$ extracted from the eICU data set. In each plot the multivariate time series is displayed. Each channel represents one vital metric which was measured from bedside sensors. Each time series covers a time span of roughly 21 hours. No obvious patterns or seasonalities can be seen.

set the daily views of 117277 articles are used. The daily views were recoreded from July 2015 to the end of 2016 over a period of 550 days. The FFT analysis indicates that the presence of seasonalities is limited. As the resolution of the data set only provides daily views and does not cover multiple years, seasonality with durations of less than a day or longer than a year cannot be observed.

As a first pre-processing step, all time series are fixed to a length of 256. Initial data analysis indicates, that the data set contains extreme outliers with in the form of peak values for daily views. The average daily views for the whole data set is 1545, while the median value is 187. These outliers have the negative effect, that a simple min-max scaling in order to normalize the data, will lead to most time series having values close to 0. This can lead to vanishing gradient problems when working with sigmoid layers for the generator. To counteract this issue, we apply a logarithmic scaling to all values. After applying this scaling, the distribution does not contain such extreme outliers. A boxplot of the peak daily views for each article before and after applying the scaling is displayed in Figure 9.

After applying the logarithmic scaling, a simple min-max

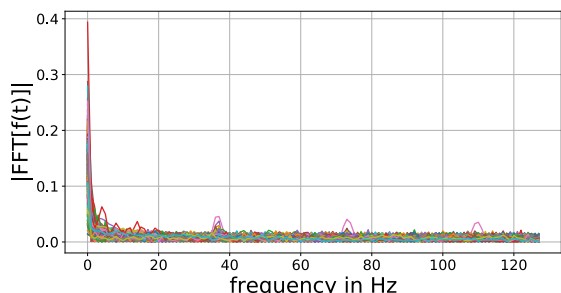

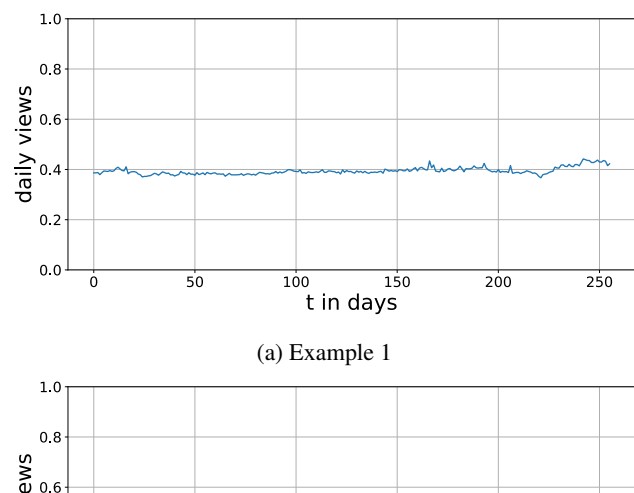

(a) Example 1

Figure 8: Result of FFT analysis applied on 1000 examples of the WWT data set. Each line represents the FFT results of the daily views of a single wikipedia article. Similar to Figure 6 no obvious patterns can be seen. The highest peak is achieved in the lowest frequency component. This frequency bin contains non-periodic components.

(b) Example 2

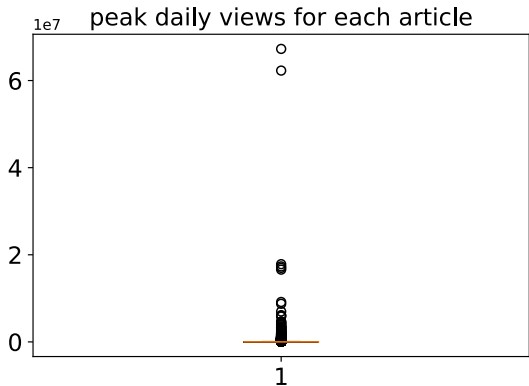

(a) Before scaling

Figure 10: Visualization of two examples from the training data set $X$ extractred from the WWT data set. Each time series covers an time span of roughly 256 days. By visual evaluation the seasonality in the WWT is lower compared to the CDN data set.

scaler is used to normalized the data set. The resulting training data set can then be defined as $X \in [0; 1]^{117277 \times 256 \times 1}$.

Examples of this training set $X$ are displayed in Figure 10.

## 5.5 Data Set Comparison

**Temporal Correlations** To quantify the findings of the FFT, we further use a time series decomposition.

The time series decomposition is not applied to the sine data set, as the sine wave is a periodical signal, with no trend or residual component. As a first step, it has to be determined, if a multiplicative or additive model is present for each data set. Considering the CDN data set, the amplitudes of the seasonal variations are not constant. Further, we could not verify the presence of any seasonalities for the eICU and WWT data set. Hence, for all data, we assume a multiplicative model. An example of a time series decomposition on an sample of the CDN data set is shown in Figure 11.

Considering a univariate time series $Y \in [0; 1]^n$, a seasonal decomposition will result in $T \in [0; 1]^n$, $S \in [0; 1]^n$ and $E \in [0; 1]^n$. For multivariate time series we apply the seasonal decomposition on each channel. In advance, we need to define the length of the seasonal component under inves-

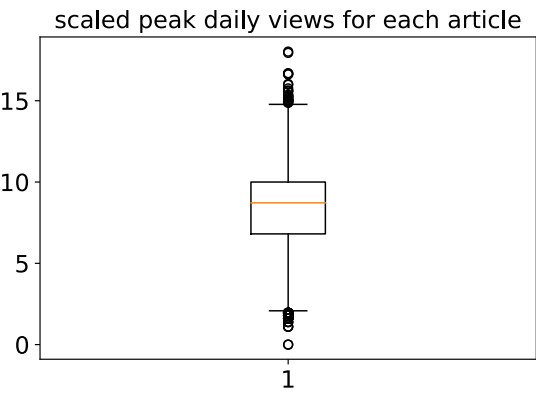

(b) After scaling

Figure 9: Visualization of the distribution of the peak daily views of the WWT data set **(a)** before and **(b)** after applying a log scaling. The data set before normalization contains extreme outliers.

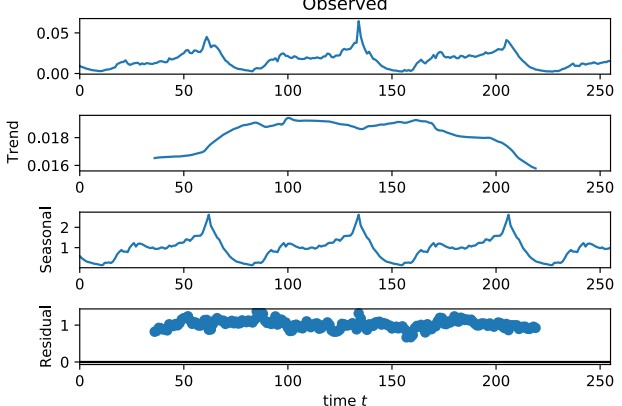

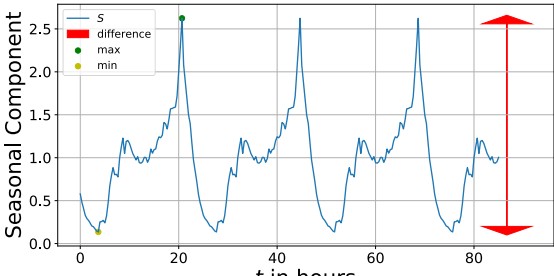

Figure 12: Visualization of max and min values of seasonal component $S$ extracted by the time series decomposition on a CDN example with a period length of a single day. The max and min values are used to calculate $r$.

Figure 11: Time series decomposition on a sample of theCDN data set. The time series decomposition is done with a period length of 72, which equals to the daily seasonalities being analyzed. The trend and residual component lack 36 values at the beginning and end of the time series due to the time series decomposition. A strong daily cycle can be observed in the seasonal component.

only cover 21 hours, seasonalities with a length of 10 minutes, one and two hours are analyzed.

The results of the time series decomposition of the CDN, WWT and eICU are listed in Table 1. The CDN data set contains strong daily seasonalities. In contrast, there are no major seasonal patterns present in the WWT or eICU data set.

tigation. Then, the influence of this seasonal component $S$ can be analyzed. For this task, we calculate the ratio between the maximum value and minimum value of the time series $S$. This ratio value indicates how big the seasonal component is in a multiplicative model, i.e, how much variance of the time series values is explained by the seasonal component. The seasonal effects in a multiplicative model can be seen as a percentage scaling factor at each time step. We utilize this percentage scaling factor to investigate the influence of the seasonal components within each time series. This allows us to compare the temporal correlations of different data sets. The procedure of calculating this ratio value $r$ for an univariate time series $Y$ with a given period length $pl$ can be noted as:

$$T, S, E = \text{seasonalDecompose } (Y, pl) \quad (5)$$

$$r = \frac{\max(S)}{\min(S)} \quad (6)$$

The calculation of this ratio value for a daily seasonality for one time series is visualized in Figure 12, where $r$ is calculated for one location of a CDN sample. In the plot, the red arrow indicates the difference between the max and min value, which are used for the $r$ calculation.

We use the median value of the calculated ratio values of all time series in a data set $X$ to calculate an influence factor $r_{pl}$. We investigate the influence of hourly and daily seasonalities in the CDN data set, with the given resolution and time series length. We consider weekly and monthly seasonalities for the WWT data set. As the used time series of the eICU data set

Table 1: Result of the seasonal decomposition of the time series extracted from the CDN, eICU and WWT data set. The $r$ values indicate the influence of the seasonal components on the time series for specific time periods in the data, ranging from ten minutes to monthly spans. A higher value indicates a higher influence.

| Data set | $r_{10min}$ | $r_{hourly}$ | $r_{2h}$ | $r_{daily}$ | $r_{weekly}$ | $r_{monthly}$ |
|---|---|---|---|---|---|---|
| CDN#0 | - | 1.0288 | 1.0784 | 22.0307 | - | - |
| CDN#1 | - | 1.0190 | 1.0631 | 19.5571 | - | - |
| CDN#2 | - | 1.025 | 1.0793 | 21.0257 | - | - |
| WWT | - | - | - | - | 1.0602 | 1.0751 |
| eICU O2 | 1.0005 | 1.0059 | 1.0118 | - | - | - |
| eICU HR | 1.0014 | 1.0212 | 1.0434 | - | - | - |
| eICU RR | 1.0066 | 1.0791 | 1.1577 | - | - | - |

**Spatial Correlations** In order to characterise the data sets with regard to spatial correlation, we utilize the Pearson correlation coefficient. For this task, we calculate the correlation coefficient between each channel for each time series of the multivariate data sets. After calculating the correlation coefficient for each time series, the average value is computed for each data set. The results of this analysis are listed in Table 2.

Table 2: Result of the spatial correlation analysis of the time series extracted from the CDN, eICU and sine data set. Values indicate the average correlation between the channels of the multivariate time series. The correlation coefficient is limited to a range between $-1$ and $1$. $-1$ and $1$ indicate a perfect correlation, while $0$ indicates no correlation between the channels of the time series.

| Data set | Average Pearson-Correlation |
|---|---|
| Sine | 1.0 |
| CDN | 0.92 |
| eICU | 0.35 |

The analysis shows that the CDN data set contains a strong spatial correlation indicated by a high Pearson correlation coefficient between the different inner-nodes. This means that the amount of user request at each inner-node correlate strongly and an increase of user requests at one location leads correlates with user requests at the other locations as well. In comparison, the multivariate time series of the eICU data set do not contain as strong spatial correlations. As to be expected, the constructed sine data set contains a perfect spatial correlation, due to the second channel of the time series merely being shifted.

## 5.6 Summary

We use four different data sets to answer the posed research question. The CDN data set contains multivariate time series with strong daily seasonality and a high spatial correlation. In contrast, the multivariate eICU data set does not contain any seasonalities and the spatial correlation is smaller. To evaluate the performance in a univariate context, we utilize the WWT data set. It does not contain a strong seasonality in the training set.

Table 3 summarizes the data sets and their characteristics.

Noticeably, even though the CDN and WWT data set are both represent user requests, they have different characteristics with regard to seasonalities and temporal correlations. This can, however, be mainly caused by the resolution of the data.

Table 3: Summary of the data sets after pre-processing, which are used for the architecture comparison.

| Data set | # examples | periodic | # channels | spatial correlation |
|---|---|---|---|---|
| CDN | 19472 | strong daily | 3 | high |
| WWT | 117277 | low | 1 | - |
| eICU | 7917 | low | 3 | low |
| Sine | 20000 | yes | 2 | perfect |

## 6 Approach

This section focuses on our approach, used to implement the methodology proposed in Section 4. We describe the architectures used for the different GAN models in detail. Afterwards, we list the proposed evaluation metrics, which we use to evaluate the generated data and GAN architectures. Lastly, we provide details about our parameter search.

### 6.1 Network Architectures

**Recurrent Neural Networks** For the RNN-based models, we utilize an LSTM block in the network architectures, as it is a common approach to counteract the limitations of vanishing gradients in RNN-based models. For the discriminator architecture, the output of the LSTM blocks is flattened and then transformed by multiple dense layers. A softmax layer is used to achieve the classification task. Initial experiments showed, that a naive LSTM block suffered from vanishing gradient problems. To counteract this issue, a skip connection is implemented over the LSTM block of the discriminator, as suggested by [26]. A batch-normalization layer [16] is used to achieve more stable training. To counteract mode collapse issues a dropout setting was used for the LSTM neurons to improve the training by introducing noise. An illustration of the discriminator architecture is displayed in Figure 13.

For the RNN generator part, we also utilize an LSTM block. Here, a naive LSTM block is sufficient as the generator did not suffer from vanishing gradient problems. This LSTM block with multiple layers and a predefined number of hidden neurons processes a sampled noise vector. In line with the discriminator part, multiple dense layers are utilized to achieve the required output dimensionality. We normalized the data sets as suggested by Lin et al. [27], and use a sigmoid layer to transform the generated time series in a range of $[0; 1]$. An illustration of this architecture can be seen in Figure 14.

**Temporal Convolutional Networks** For the TCN-based models, we utilize multiple temporal blocks as proposed by Bai, Kolter, and Koltun [3]. The temporal blocks are used to process the input time series.

The discriminator uses a temporal block to classify the input time series. In line with Wiese et al. [36] we use the last element of each feature map, named head element. This head element contains all necessary information if the receptive field size of the TCN block covers the whole input sequence. If this is not satisfied an information loss in the early parts of the time series is occurring. As with the RNN-based architecture, we use multiple dense layers to achieve the required output dimensionality. The output is fed through a softmax layer to calcuate the classification scores.

We reuse the suggested discriminator architecture for the generative model and only modify the dense layers for this

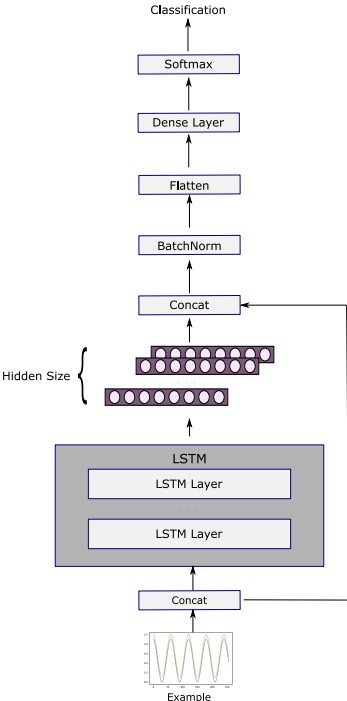

Figure 13: LSTM based discriminator architecture. The sample, which should be classified is fed into multiple LSTM-layers with a fixed number of hidden neurons. A skip connection over the LSTM block is used to counteract the problem of vanishing gradients. A batch normalization layer [16] is used to achieve more stable training. The transformed time series is then flattened and fed into a dense layer. Afterwards a softmax or sigmoid layer is used to achieve the classification task.

task. An illustration of the architectures can be found in Figure 15 and Figure 16.

**Transformer**    We use a transformer architecture based on Lim et al. [26]. The authors utilized this TFT for a time series prediction task. For this task, the they feed static information, observed, past and known future information into the TFT model. We use their TFT as a basis and modify it for our data generation task. In a GAN context, no static information or known future information is present, we therefore remove this components of the architecture. The attention block within the TFT architecture uses a masked multi-head attention mechanism, which processes the past inputs and known future inputs. We replace this multi-head attention mechanism with a multi-head self attention mechanism, which solely processes the time series (corresponding to past inputs in a multi horizon context). In contrast to the current state of the art, our transformer architecture is specifically modified for the data generation task. The exact architecture for discriminator and generator model is as follows: The inputs to the modified

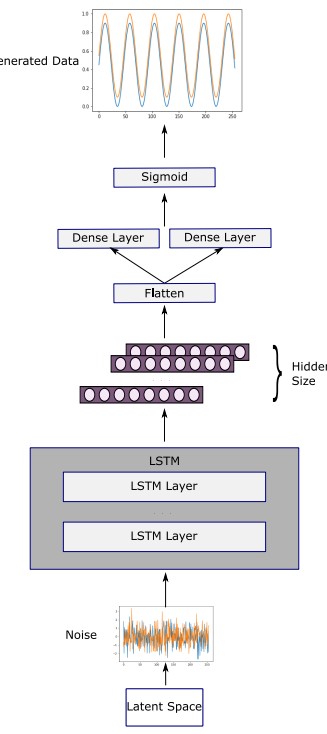

Figure 14: LSTM based generator architecture. The sampled noise vector is fed into multiple LSTM layers. The output of the LSTM layers is flattened and multiple dense layers are applied to achieve the mutlivariate time series. A sigmoid layer is used to transform the generated time series into a range of $[0, 1]$.

TFT architecture are fed into an embedding layer realized by a dense layer. After this embedding a Variable Selection Network is applied. The Variable Selection Network is used for judicious selection of the multivariate input vector [26]. The Variable Selection Network is realized by applying gated residual networks, which can be seen as a dynamic skip connection mechanism [14]. A positional encoding is added to the time series before feeding it into an LSTM block. The LSTM block is realized by multiple LSTM layers with a fixed number of hidden neurons. The output of the LSTM block is combined with a skip connection and fed into a gating component realized by a gated linear unit [8] in order to focus on the important features of the time series. A batch normalization layer is used to increase the stability of the training. Then, a multi-head self attention mechanism is used to transform the time series with an unlimited look back. The output of this transformer component is combined with a skip connection. Afterwards multiple gating and normalization components are applied. Lastly, multiple dense layers are used to achieve the desired output dimensionality for discriminator and generator accordingly. In the discriminator model a softmax layer is utilized to achieve the classification scores, while the generator

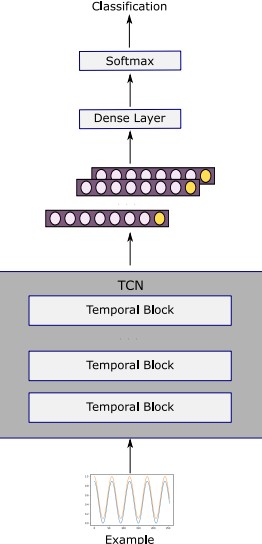

Figure 15: TCN based discriminator architecture. The sample, which should be classified is fed into multiple temporal blocks. Afterwards the heads of the transformed time series are used and a single dense layer is applied to achieve a vector with the desired number of classes. Lastly, a softmax layer is used to classify the sample.

uses a sigmoid layer.

While we reduce the complexity of the original model [26] due to smaller number of input features, the complexity of the architecture, specifically when compared to the RNN and TCN setup is evident.

Illustrations of our modified TFT architecture are displayed in Figure 17 and Figure 18.

## 6.2 Evaluation Metrics

In order to evaluate the performance of a GAN model, the underlying ground truth data distribution has to be compared with the estimated distribution. When comparing two unknown distribution, only samples drawn from these distributions are available. In the case of a GAN, sampling from the learned distribution corresponds to transforming noise vectors into time series with the generator. To evaluate how successful the estimation of the probability distribution is, multiple empirical metrics are utilized. We focus on evaluating the performance by considering the similarity of the distributions with regard to temporal correlations, spatial correlations, noise and mode collapse issues.

### 6.2.1 Squared Difference Calculation

We compute different evaluation metrics ($M$) on a the generated time series and the training set. A similar approach was proposed by Lucic et al. [28], which evaluated generated and real samples every five epochs. The chosen training time

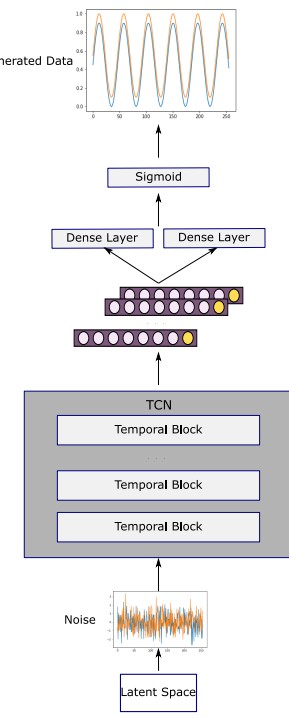

Figure 16: TCN based generator architecture. The sampled noise vector is fed into multiple temporal blocks. Afterwards, the heads of the transformed time series are used and multiple dense layers are applied to achieve the mutlivariate time series. A sigmoid layer is used to transform the generated time series into a range of $[0, 1]$.

series to be compared are selected at random. We define the set of generated time series as $G$, and the set of chosen training time series as $R$. We calculate the metrics of $M$ on each time series within each set. Before training the GAN models, we define a fixed noise vector in advance. Fixing this noise vector helps analyze the progress and convergence of the GAN model. This defined noise vector is used to generate the set $G$ at each training epoch. We define this noise vector to generate ten time series at each training epoch to build the set $G$ ($|G| = 10$) .

Before training a GAN model, we sample 500 random time series from the training data to create set $R$ ($|R| = 500$). In the following we will note the set $G$ as the "generated" set and the set $R$ as the "reference" set. To empirically measure how similar set $G$ and $R$ are with regard to a defined metric, we compute the average value within each set and calculate the squared difference between both values. The process of this squared difference calculation used to account for temporal correlations, spatial correlations and entropy is visualized in Figure 19.

Generally speaking, considering an evaluation metric $f(x)$ with:

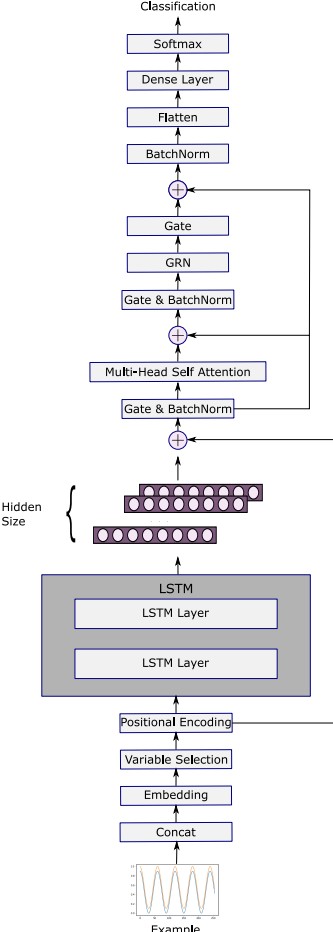

Figure 17: Transformer based discriminator architecture. The sample is fed into an embedding dense layer and afterwards a variable selection network realized by gated residual networks [14]. An LSTM block is used to detect temporal correlations in the time series. Afterwards, multiple gating components are used to focus on the important parts of the time series. A multi-head self attention mechanism is used to achieve an unlimited receptive field size. After applying additional gating and normalization layers, the flattened time series is fed into a dense and sigmoid or softmax layer to achieve the classification task.

$$f(x) : [0;1]^{256 \times d} \rightarrow \mathbb{R} \qquad (7)$$

with $d$ being the dimensionality of the time series, which transforms a time series into an empirical value, the calculation can be described as:

$$f^{ch}_{reference} = \frac{\sum_{r_j \in R} f(r_j)}{|R|}, 1 \le ch \le d \qquad (8)$$

$$f^{ch}_{generated} = \frac{\sum_{x_j \in G} f(x_j)}{|G|}, 1 \le ch \le d \qquad (9)$$

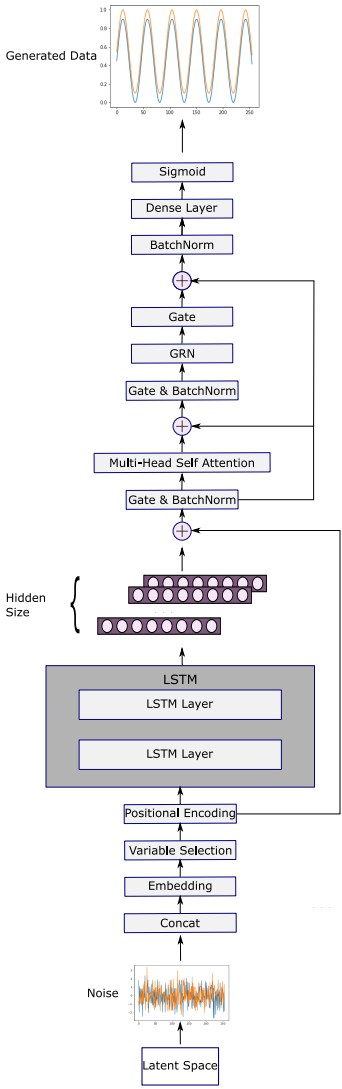

Figure 18: Transformer based generator architecture. The noise is fed into an embedding dense layer and afterwards a variable selection network realized by gated residual networks [14]. A LSTM block is used to detect temporal correlations in the time series. Afterwards, multiple gating components are used to focus on the important parts of the time series. A multi-head self attention mechanism is used to achieve an unlimited receptive field size. After applying additional gating and normalization layers a dense and sigmoid layer is utilized to generate the desired time series.

$$f_{output} = \frac{\sum_{ch=1}^{d} \left( f^{ch}_{reference} - f^{ch}_{generated} \right)^2}{d} \qquad (10)$$

With our approach, this $f_{output}$ value indicates how similar $R$ and $G$ are with regard to the considered evaluation metric. The value specifies the difference between the set $G$ and $R$, therefore a smaller value corresponds to a better data

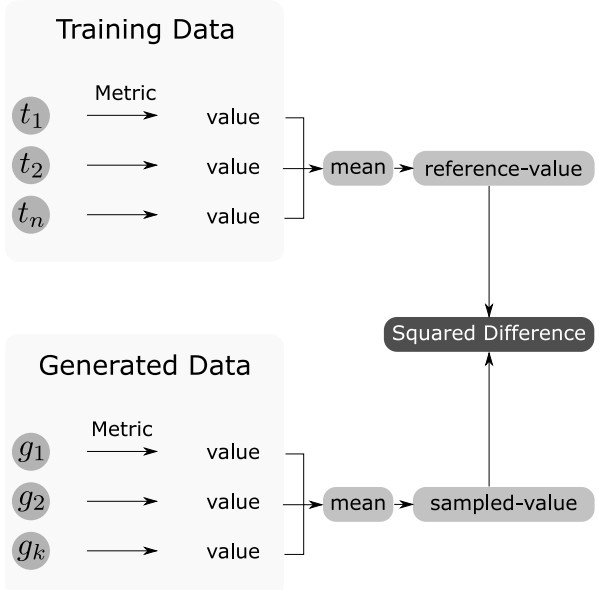

Figure 19: Visualization of the calculation of the squared difference used to compare two distribution based on a predefined metric. $t_i$ represents a time series drawn from the training set, with $n = 500$, while $g_j$ represents a generated time series, with $k = 10$. This approach is used for Discrete Fourier Transformation (DFT), Pearson correlation and entropy.

distribution estimation by the GAN model.

We consider following metrics for the squared difference calculation:

**Temporal Correlation**  Common approaches compute a DFT or an auto-correlation for to account for temporal correlations. While Lin et al. [27] utilized the auto-correlation to analyze the temporal correlation, we propose to leverage the DFT. The information gained by an auto-correlation analysis is similar to an DFT as it represents the normalized spectral density after a DFT [21]. Our initial experiments indicate, that undesired noise included in generated time series can be more easily detected by a DFT than an auto-correlation analysis. To visualize this pattern, Figure 20 compares the output of an FFT and auto-correlation computed on sine waves with different amount of noise.

We use an FFT analysis to transform a time series into its frequency domain. By dividing each amplitude value by the sum of all amplitudes in the frequency spectrum, we compare the distributions of frequencies. With this in mind, we remove the lowest frequency bin to discard the frequency components extracted from non-periodic components of the time series. For our comparison, we only compare the highest probabilities (peaks) of the frequency spectrum. As a result of data analysis for the training data sets the number

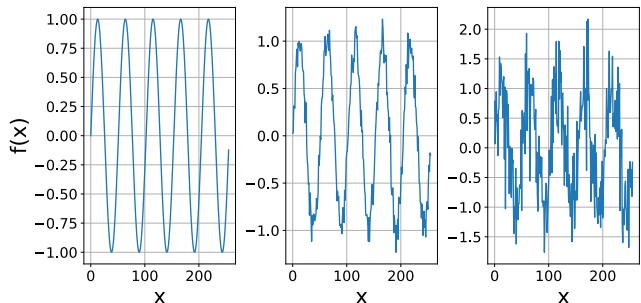

(a) Sine Data with varying noise sampled from a normal distribution.

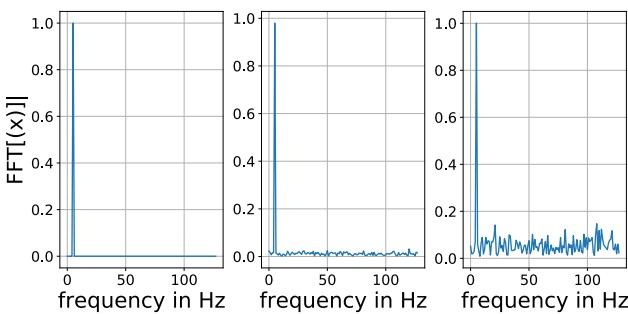

(b) Frequency domain of sine waves from Figure 20a achieved by applying FFT. Added noise can also be found in frequency domain.

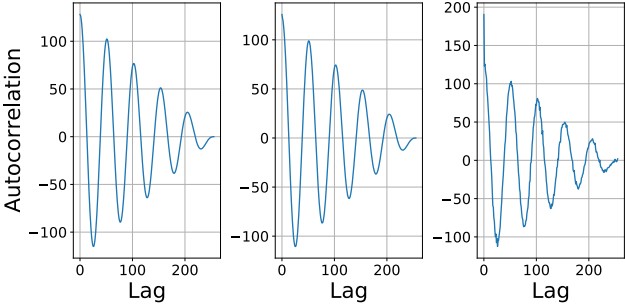

(c) Auto-correlation applied on sine waves from Figure 20a.

Figure 20: Illustration of FFT and auto-correlation calculation on sine data with varying noise. In columns from left to right more noise is added to the signal. Compared to FFT, the added noise is harder to detect by the auto-correlation analysis.

of relevant frequencies peaks, which should be compared are pre-determined. All time series within the sine and cosine data consist of a single frequency component. Therefore, we consider only the frequency with the highest amplitude. For the CDN data set analysis has shown, that the time series consists of three major frequency components. For the eICU and WWT data set, only the frequency with the highest amplitude is considered. The amplitudes of the extracted frequency components are then compared and the distance is calculated, which should be as small as possible. Since we only aim to account for temporal correlations with this measurement, this

FFT comparison is done for each channel in the time series separately.

**Spatial Correlation**  We follow the approach suggested by Leznik et al. [24] to utilize a correlation coefficient to measure the spatial correlation within the generated time series. For the correlation coefficient we compute the Pearson correlation. We consider all possible channel tuples for the calculation of $f_{output}$. If the generated samples are drawn from the same distribution a similar correlation coefficient should be found.

**Noise and Uncertainty**  Following the approach of Leznik et al. [24] we calculate and compute an entropy measurement. With such a measurement we can evaluate if the generated time series contain similar noise and uncertainty information. Here, the Approximate Entropy (ApEn), which is an adaption of the entropy measurement for time series, is used, as it allows to measure the repeatability and predictability within a time series.

**Limitations**  The squared difference comparison based on the mentioned measurements helps to evaluate how similar both distributions are with regard to temporal correlation, spatial correlation and more. However, with our approach, the squared difference is calculated based on an average value extracted from samples of these distributions. This approach does not provide insight how the drawn samples are distributed in the multi-dimensional space. Specifically, we cannot compare distinct samples but only the average values calculated based on these samples.

### 6.2.2 Time Series Similarity

We include time series similarity measurements for our evaluation to counteract the limitations of the squared difference calculation. Measuring the similarity between time series data can help detect the nearest neighbour in the multi-dimensional space. In order to apply a similarity measurement we need to define a distance metric first. For our approach, we utilize the ED to measure the similarity between time series. For multivariate time series we average the distances computed on each channel.

Following the approach of Arnout et al. [2] we use the Incoming Nearest Neighbor Distance (INND) and Outgoing Nearest Neighbor Distance (ONND) to gain insight about the performance of GANs models by comparing the distance of samples from the underlying and estimated distribution. In addition, we compute the Intra-Class Distance (ICD) for the generated time series.

We use the INND to measure a successful estimation of the distribution function, which correlates to the model generating realistic time series. To empirically evaluate the flexibility of the generated time series we use the ONND measurement.

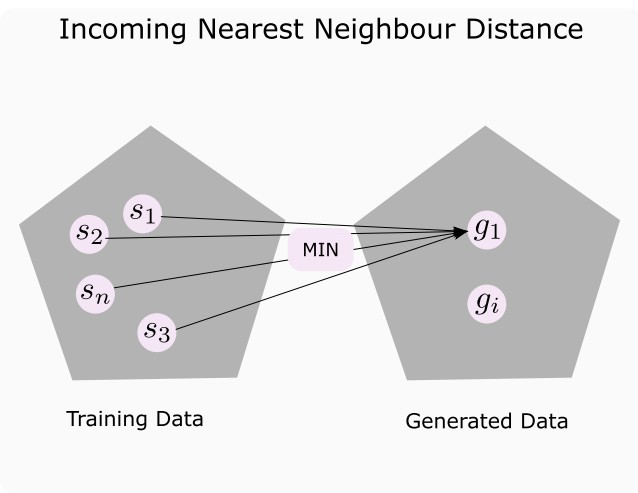

Figure 21: Illustration of Incoming Nearest Neighbour Distance. Considering training data set $R = \{s_1, s_2, s_3, \ldots, s_n\}$ and generated data $G = \{g_1, \ldots, g_i\}$ and a defined distance measurement $d(s_k, g_j) : \mathbb{R}^{n \times d} \times \mathbb{R}^{n \times d} \rightarrow \mathbb{R}^+$ the nearest neighbour of the training data set is calculated for each generated data point $g_j$.

However, the ONND metric does not consider the distances between the generated samples explicitly, hence we include the ICD measurement to account for mode collapse issues.

**Incoming Nearest Neighbour Distance**  The INND calculates the nearest neighbour of the training data set $R$ for each generated example of set $G$. An illustration of the INND based on a set of training data and set of generated data is displayed in Figure 21.

A successful estimation of the distribution function should result in close neighbours in the training data set for each generated data sample. For our evaluation, we calculate the distance to the closest neighbour in a set $R$ for each generated time series in $G$ and average these distances. Considering the set of generated samples $G$ and the set of training samples $R$, we compute the overall *innd_score* with a predefined distance measurement $d(a, b)$ as:

$$INND(x_j, R) = \min\{d(x_j, s_k) | r_k \in R\} \qquad (11)$$

$$innd\_score = \frac{\sum_{x_j \in G} INND(x_j, R)}{|G|} \qquad (12)$$

We expect that in the training process of a successful GAN the *innd_score* decreases. However, the *innd_score* should not equal 0, which would indicate that the GAN can only generate copies of the training data.

**Outgoing Nearest Neighbour Distance**  The ONND calculates for each training example of set $R$ the nearest generated

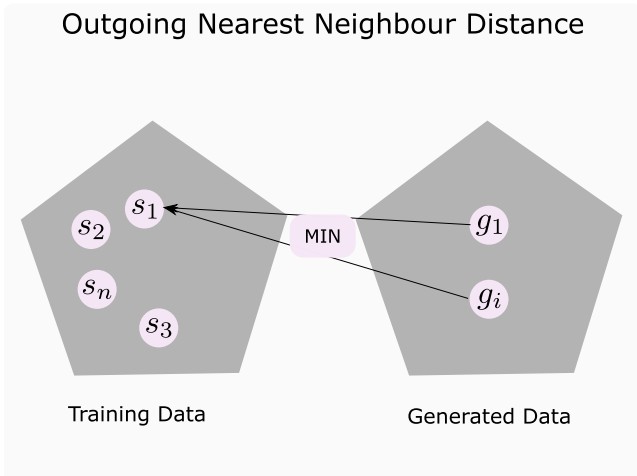

Figure 22: Illustration of Outgoing Nearest Neighbour Distance. Considering training data set $R = \{s_1, s_2, s_3, \ldots, s_n\}$ and generated data $G = \{g_1, \ldots, g_i\}$ and a defined distance measurement $d(s_k, g_j) : \mathbb{R}^{n \times d} \times \mathbb{R}^{n \times d} \to \mathbb{R}^+$ the nearest neighbour of the generated data set is calculated for each training data point $s_k$.

example in set $G$. An illustration of the ONND based on a set of training data and set of generated data is displayed in Figure 22.

Ideally, a GAN model should result in a small ONND for most training samples in set $R$. This corresponds to the GAN model being able to produce time series in the multi-dimensional space covered by the training data distribution. As with the INND measurement, we calculate the distance to the closest neighbour and average these distances. Considering the set of generated samples $G$ and the set of training samples $R$, we compute the overall *onnd_score* with a predefined distance measurement $d(a, b)$ as:

$$ONND(x_j, G) = \min\{d(x_j, g_k) | g_k \in G\} \qquad (13)$$

$$onnd\_score = \frac{\sum_{x_j \in R} ONND(x_j, G)}{|R|} \qquad (14)$$

We expect, that in a successful training process of a GAN, which is able to produce time series without mode collapse issues, the *onnd_score* decreases. The *onnd_score* provides an indication about possible mode collapse issue by measuring which areas of the multi-dimensional space are covered by the generated samples of set $G$.

**Intra Class Distance**  The ICD represents the average similarity of the generated time series within set $G$. We compute the *icd_score* considering a distance metric $d(a, b)$ and the set of generated samples $G$ as:

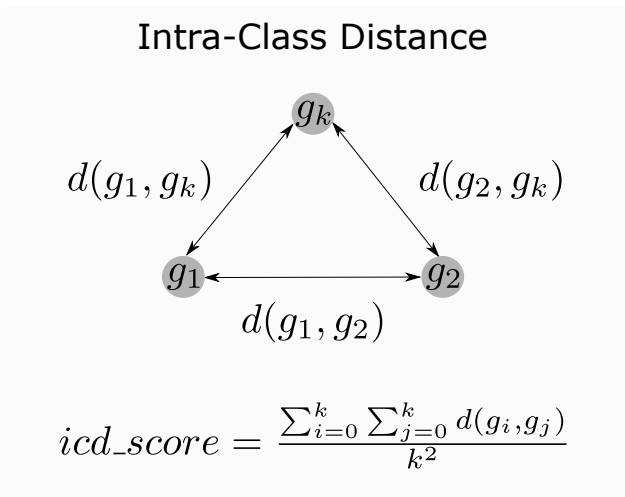

Figure 23: Illustration of Intra Class Distance. Considering the generated data $G = \{g_1, g_2, \ldots, g_k\}$ and a defined distance measurement $d(g_i, g_j) : \mathbb{R}^{n \times d} \times \mathbb{R}^{n \times d} \to \mathbb{R}^+$ the average distance between the generated examples is computed.

$$icd\_score = \frac{\sum_{x_j \in G} \sum_{x_k \in G} d(x_j, x_k)}{|G|^2} \qquad (15)$$

Figure 23 visualizes the computation of the *icd_score*.

A GAN model with mode collapse issues would result in similar samples within set $G$. In contrast to the *innd_score* and *onnd_score*, a higher *icd_score* indicates a better GAN performance.

## 6.3  Parameter Search

In order to compare the performance of the different GAN architectures, we evaluate multiple configurations to achieve a better generalization of the research results. For this task we utilize a grid search. Such a grid search was already utilized by Bai, Kolter, and Koltun [3]. While Lucic et al. [28] performed a random search, with a vast amount of varying parameters, we chose a grid search with a smaller amount of varying parameters to cover the entire search space. For our grid search we consider architecture dependent parameters as well as GAN training specific parameters. As we are utilizing a grid search, we limit the scope of the search space by fixing other parameters in advance. Based on our computation, we assume that the parameters are transferable to other data sets, as suggested by Lucic et al. [28].

### 6.3.1  LSTM Parameters

For the LSTM based models we compare the performance of LSTM blocks with one and two LSTM layers. In line with Bai, Kolter, and Koltun [3], we choose this parameter for the grid search to see if multiple LSTM layers provide

any benefit. We fixed 100 hidden neurons for the LSTM layers. This configuration was already suggested by Leznik et al. [24] and Esteban, Hyland, and Rätsch [10]. We tested other number of hidden neurons for the LSTM layer, but the preliminary results on the sine data set indicated, that the configurations proposed by the authors work best. For the dropout layer used within the LSTM block we choose a dropout of 0.2.

### 6.3.2 TCN Parameters

For the TCN based models, we compare the performance of temporal blocks with 10 and 20 channels. Following the approach of Bai, Kolter, and Koltun [3] we choose the kernel size and number of layers to result in a receptive field size, which covers to whole input sequence. We set the kernel size to 7 and 8 layers, resulting in a receptive field size bigger than 256. As the sequence length is fixed for all data sets, we use these parameters for all experiments.

### 6.3.3 TFT Parameters

In line with our grid search for the LSTM models we test if additional LSTM layers improve the performance of TFT based models. For the other parameters we chose the parameters suggested by Lim et al. [26] for the UCI Electricity Load Diagrams[5] dataset. We modify the dropout value to 0.2 based on initial results. We did not observe performance changes in those initial experiments when varying other parameters. Therefore, we used an embedding dimension of 8, 160 hidden neurons and 4 attention heads for all experiments.

### 6.3.4 GAN Parameters

We evaluate architectural independent configurations to increase the possible generalization of the results. E.g., different setups for the alternating training between the generator and discriminator are tested.

**Alternate Training**  A GAN model is optimized by alternating between training the discriminator and generator. A good balance between training these two components is required to achieve a sufficient results of the minimax game [11]. However, there is no common approach for choosing this balance. Increasing the number of times the generator is trained in comparison to the discriminator helps to improve the convergence of the GAN training. Yet, training the generator too often can lead to a mode collapse, where the generator only produces one example, which is classified as most plausible by the discriminator. We vary the alternating training for our comparison task by training the generator and discriminator equally often or training one component three times more

---

often than the other one. Lucic et al. [28] considered a similar setting for their comparison.

**Training Details**  For our experiments we use Adam optimizers [22] with a learning rate of 0.00005, beta1 of 0.5 and beta2 of 0.999 to train the discriminator and generator components. The Adam optimizer includes an momentum term for its gradient descent to speed up the training procedure. In the initial tests, we considered different learning rates and an RMSProp optimizer for comparison reasons. Based on the initial experiments on the sine data set we chose the aforementioned configurations.

For the loss function we utilize the modified minimax loss function. We also modified our GAN models to use the Wasserstein loss proposed by Arjovsky, Chintala, and Bottou [1]. We considered a clipping approach and gradient penalty approach to enforce the weight constraints. For the clipping approach we noticed, that the clipping parameters suggested by Arjovsky, Chintala, and Bottou [1] are not transferable to time series architectures. With the gradient penalty approach, we only observed an increase in training time. Therefore, we use the modified minimax loss function for all experiments.

We sample our noise vectors for the generator from a standard normal distribution. A standard normal distribution was already utilized by Esteban, Hyland, and Rätsch [10] and Leznik et al. [24]. We selected the noise vector to be of the same dimensionality as the time series in the training data set. In the initial experiments we tested if an increased dimensionality would help, but we could not verify this assumption.

A batch size of 512 is used. This setting resulted in the best trade-off between performance and training time in the initial experiments compared to lower batch sizes.

### 6.3.5 Procedure

We consider mixed architectures, based on TCN, LSTM and TFT for the comparison task. The set of architectures $A$ can be defined as:

$$A = \{TCN, LSTM, TFT\} \times \{TCN, LSTM, TFT\} \quad (16)$$

Resulting in nine different generator and discriminator combinations. We perform the parameter search for each architecture individually. To reduce computation time, the parameter search is conducted on the artificial (sine) and one real (CDN) data sets. The parameters of the CDN data set are then transferred to the WWT and eICU data sets.

We then compare the architectures based on the best performing configurations. Here we evaluate performance on all data sets. As suggested Lucic et al. [28] we train additional models for the best performing configuration to account for

---

[5]https://archive.ics.uci.edu/ml/datasets/ElectricityLoadDiagrams20112014

instability issues. For this task we train 5 additional models for all data sets for each architecture and average their measurement scores.

## 6.4 Inference Process

While generating time series data for evaluation purposes, we do not deactivate the dropout layers in the generator models. While this is uncommon for standard ANN tasks this approach was already utilized by Isola et al.[17]. In a GAN context this helps to generate time series with high fidelity and counteract mode collapse issues by introducing noise to the generation process.

## 7 Results

In the following, we provide the empirical results of the comparison runs between the different architectures. Based on the combination of different GAN architectures and the parameter space, our experiments contain 612 trained GAN models, with an overall computation run time of 2610 hours.

### 7.1 Parameter Search

To allow a fair comparison, we conducted an extensive parameter search to find the optimal setup for each GAN architecture. In the following we will refer to $(G, D)$ as a GAN model. $G$ and $D$ are placeholders for the corresponding generator and discriminator architectures. The results of the parameter search are listed Table 4.

### 7.2 Quantitative Results

In the narrowed search, we train five GAN models of the best performing configurations for each $(G, D)$ combination. The scores for each evaluation metric were averaged over these five runs. The results for all data sets in Table 5. The values are rounded to five decimal places and the three best performing architectures according to each metric are marked bold.

**Sine & Cosine**  Considering our evaluation metrics, the architectures with a TCN discriminator performed the best. A full TCN setup, with a TCN generator and discriminator part had the most successful data synthesis over all runs. As the frequency component solely defines the training data, our temporal correlation metric can give the best indication of a successful data distribution estimation. When analysing this metric it can be seen, that the setups with a TCN discriminator are the most stable, which correlates to our other measurements. We use the ICD to evaluate the performance with regard to a possible mode collapse. A low score in this metric indicates that a mode collapse occurred. Table 5 shows, that a GAN setup with a TCN generator and LSTM discriminator

Table 4: Results of the parameter search. The best performing configurations for each $(G, D)$ GAN. The columns indicate which parameters were varied for the parameter search. The specific search is explained in Section 6.3

| TFT, TFT | | | | |
|---|---|---|---|---|
| Data Set | D Steps | G Steps | G # layers | D # layers |
| artificial | 3 | 1 | 2 | 1 |
| real | 1 | 3 | 1 | 2 |

| TCN, TFT | | | | |
|---|---|---|---|---|
| Data Set | D Steps | G Steps | G # channels | D # layers |
| artificial | 3 | 3 | 10 | 2 |
| real | 1 | 1 | 10 | 2 |

| LSTM, TFT | | | | |
|---|---|---|---|---|
| Data Set | D Steps | G Steps | G # layers | D # layers |
| artificial | 1 | 3 | 1 | 1 |
| real | 3 | 3 | 1 | 1 |

| TFT, TCN | | | | |
|---|---|---|---|---|
| Data Set | D Steps | G Steps | G # layers | D # channels |
| artificial | 3 | 3 | 2 | 10 |
| real | 3 | 1 | 2 | 10 |

| TCN, TCN | | | | |
|---|---|---|---|---|
| Data Set | D Steps | G Steps | G # channels | D # channels |
| artificial | 3 | 3 | 10 | 20 |
| real | 1 | 1 | 10 | 20 |

| LSTM, TCN | | | | |
|---|---|---|---|---|
| Data Set | D Steps | G Steps | G # layers | G # channels |
| artificial | 3 | 1 | 1 | 20 |
| real | 3 | 1 | 1 | 20 |

| LSTM, LSTM | | | | |
|---|---|---|---|---|
| Data Set | D Steps | G Steps | G # layers | D # layers |
| artificial | 3 | 1 | 2 | 1 |
| real | 3 | 1 | 1 | 1 |

| TCN, LSTM | | | | |
|---|---|---|---|---|
| Data Set | D Steps | G Steps | G # channels | D # layers |
| artificial | 1 | 3 | 20 | 2 |
| real | 1 | 1 | 10 | 1 |

| TFT, LSTM | | | | |
|---|---|---|---|---|
| Data Set | D Steps | G Steps | G # layers | D # layers |
| artificial | 3 | 1 | 1 | 2 |
| real | 3 | 3 | 1 | 1 |

| Generator | Discriminator | Temporal Correlation | Spatial Correlation | ICD | INND | ONND | Appx. Entropy | Temporal Correlation | Spatial Correlation | ICD | INND | ONND | Appx. Entropy |
|---|---|---|---|---|---|---|---|---|---|---|---|---|---|
| | | Sine | | | | | | eICU | | | | | |
| TCN | TCN | **0.02332** | **0.00015** | **10.06231** | **0.53573** | **72.74083** | **0.00148** | 0.00200 | 0.10909 | 2.46944 | **1.67306** | **3.50805** | 0.08154 |
| LSTM | TCN | **0.01996** | 0.00075 | 1.77302 | **9.67138** | 88.52687 | **0.00317** | 0.00261 | 0.075323 | 0.78862 | 2.73161 | 4.94537 | **0.07222** |
| TFT | TCN | **0.01775** | **0.00000** | 1.05250 | **12.23117** | **80.75530** | **0.00926** | **0.00183** | 0.405765 | **29.29440** | 16.44228 | 7.51501 | **0.04232** |
| TCN | LSTM | 0.07313 | 0.15511 | **7.12300** | 58.69233 | 111.73984 | 0.17102 | 0.00395 | **0.01481** | 0.12036 | 139.37538 | 153.68854 | 0.08339 |
| LSTM | LSTM | 0.10324 | 0.17809 | 0.03923 | 96.70890 | 153.44000 | 0.09419 | 0.00337 | **0.02525** | 0.00881 | 181.20906 | 222.09209 | 0.09554 |
| TFT | LSTM | 0.04116 | **0.00040** | **18.57820** | 34.09213 | **79.64032** | 0.05801 | 0.00325 | 0.23396 | 7.70013 | **2.72719** | **3.14137** | **0.07647** |
| TCN | TFT | 0.10878 | 0.50436 | 1.26913 | 94.31486 | 120.93548 | 0.09099 | **0.00167** | 0.06978 | **21.92763** | 43.52272 | 48.16701 | 0.08618 |
| LSTM | TFT | 0.09896 | 0.56957 | 0.00797 | 74.99265 | 101.63118 | 0.25078 | 0.00236 | 0.06175 | **16.83475** | 31.82086 | 43.68840 | 0.17196 |
| TFT | TFT | 0.08530 | 0.34764 | 1.38502 | 68.03877 | 93.94007 | 0.11558 | **0.00161** | **0.04572** | 4.04906 | **0.66680** | **2.35166** | **0.05636** |
| | | CDN | | | | | | WWT | | | | | |
| TCN | TCN | **0.00055** | **0.00374** | 10.2777 | **3.17988** | **6.59869** | **0.015180** | **0.00028** | | 7.03195 | 0.35826 | 1.07870 | **0.00080** |
| LSTM | TCN | **0.00038** | **0.00381** | **14.94560** | **3.12956** | **5.98848** | **0.00526** | **0.00020** | | 7.00606 | **0.32634** | 0.99887 | **0.00236** |
| TFT | TCN | 0.00106 | 0.00587 | **44.44957** | 38.61944 | 21.82845 | 0.02415 | **0.00022** | | **7.70210** | **0.30861** | **0.87396** | **0.00186** |
| TCN | LSTM | 0.00842 | 0.62799 | 0.36461 | 137.72078 | 163.00755 | **0.00813** | 0.00182 | | **7.95031** | 55.20507 | 61.08195 | 0.13351 |
| LSTM | LSTM | 0.00595 | 0.56521 | 0.00456 | 148.11764 | 199.05331 | 0.08650 | 0.00112 | | 0.01699 | 28.05824 | 45.40631 | 0.13530 |
| TFT | LSTM | **0.00088** | **0.00133** | 12.85012 | **5.18976** | **7.14789** | 0.03194 | 0.00085 | | 7.44699 | **0.27199** | **0.85406** | 0.01187 |
| TCN | TFT | 0.00370 | 0.36310 | **24.79750** | 84.80797 | 93.53913 | 0.03446 | 0.00063 | | **8.36950** | 3.08563 | 6.51213 | 0.09195 |
| LSTM | TFT | 0.00239 | 0.195973 | 4.03003 | 65.44299 | 97.71488 | 0.03276 | 0.00165 | | 3.12135 | 11.49566 | 29.11385 | 0.57385 |
| TFT | TFT | 0.00203 | 0.62835 | 14.38590 | 25.20930 | 25.82004 | 0.03994 | 0.00035 | | 7.22410 | 0.33559 | **0.93286** | 0.00236 |

Table 5: Evaluation metrics of the different GAN architectures for all four data sets. For each metric, the best three scores are marked as bold.

as well as a GAN setup with a TFT generator and LSTM discriminator resulted in high ICD values. However, these models were not able to recover the data distributions successfully and only produced noise. When these outliers are not considered, a full TCN setup was the only model without mode collapse issues for the artificial data set.

**CDN** A before, the architectures with a TCN discriminator performed best. For the CDN data set a setup with an LSTM generator and a TCN discriminator had the most successful data synthesis over all runs. When considering the INND measurement, which indicates the fidelity of the generated data, it is noticeable that architectures without a TCN discriminator struggle to synthesise realistic time series. This pattern is also indicated by our temporal correlation measurement, which is important for the CDN data set with high seasonalities. The synthesis of the CDN data set resulted in fewer mode collapse issues. Most architectures were able to generate examples with high variance for the CDN data set, which is shown by higher ICD values. Noticeably, the synthesis of rare events does not deviate the behaviour of the GANs. We attribute this to the fact that the point and collective anomalies are averaged out while calculating the data statistics. However, the goal here is to show the ability to mimic the distribution including the anomalous data points.

**eICU** For the eICU data set, a setup with a TFT generator and a TFT discriminator had the most successful data synthesis over all runs. This is shown by a high fidelity according to all evaluation metrics. While GAN models with a TFT discriminator and without a TFT generator performed worse. It is generally noticeable, that architectures with a TCN discrim-

inator offer sufficient all around performance. The training of the GAN models was stable with regard to mode collapse issues. According to the INND metric, it can be seen that in addition to the TCN discriminator based models, the TFT based models are able to generate realistic examples.

**WWT** Architectures with a TCN discriminator performed the best and the estimated data distribution of a full TFT GAN is similar to the training data distribution. While the full TFT GAN did not achieve top scores in most measurements, a high fidelity of the generated data is indicated by stable scores in all metrics. For the WWT data set, a setup with a TFT generator and a TCN discriminator had the most successful data synthesis over all runs. Considering the ICD of the generated examples, nearly all architectures were able to estimate data distributions with no mode collapse. When analyzing the INND, all TCN-discriminator based models were able to produce samples, which have close neighbours in the training data distribution.

## 7.3 Qualitative Results

We provide qualitative analysis in this section, first, we use the qualitative analysis to visualize the issues of mode collapse for the sine data set. Then, we utilize verify the fidelity of the generated time series. For this task we provide a generated time series for each data set, its closest neighbour within the training data set, and a random time series from the training data set.

### 7.3.1 Mode Collapse Issues

For visualization purposes, Figure 24 shows the output of two GAN models. The GAN models were trained to synthesise the on the sine data set. One GAN model is successful, while the other one suffered mode collapse issues. The following figure displays this pattern, which was already empirically captured by the ICD (see Table 5). While the sine waves seem alike at first glance, notice how the frequency of the example samples does not vary at all in the mode collapse case, while the successful training examples differ in their frequency.

### 7.3.2 Fidelity of Generated Time Series

After visualizing generated samples for our artificial multi-class data set in Figure 24, we provide visual examples of generated time series for the real data sets. The synthesised time series are displayed in Figure 25.

The figure shows synthesised examples for each data set, with their nearest neighbour and a random time series of the corresponding data set. We use the ED to detect the nearest neighbour. The provided generated time series were synthesised by full TCN GANs, as they resulted in the best overall data estimation for all data sets.

**Sine & Cosine**  The TCN GAN was able to produce realistic time series for the exemplary artificial data set. In addition, the generated time series are well behaved periodic waves with close to zero noise.

**CDN**  The time series synthesised for the CDN data set has similar characteristics with regard to cycles and spatial correlations to the training data. The GAN was able to produce a time series with diurnal cycles and high spatial correlations.

**CDN - Rare Events**  The synthesis of anomalous events shows, depicted in Figure 26, shows the ability to synthesize the data distribution, including anomalous events. We opted to show the rather easily distinguishable case of a collective data anomaly in the form of a simulated outage. However, given the ability to replicate the underlying data distribution, the argument for the synthesis of more subtle rare and anomalous events such as change points can be made.

**eICU**  The synthesised time series displayed in Figure 25b is similar to its nearest neighbour. This indicates high fidelity of the generated time series. However, the generated time series is not an exact copy of the training data, which is desired for time series synthesis.

**WWT**  The generated time series displayed in Figure 25c has a close neighbour within the training set in the multi-dimensional space. In contrast, the random time series, cap-

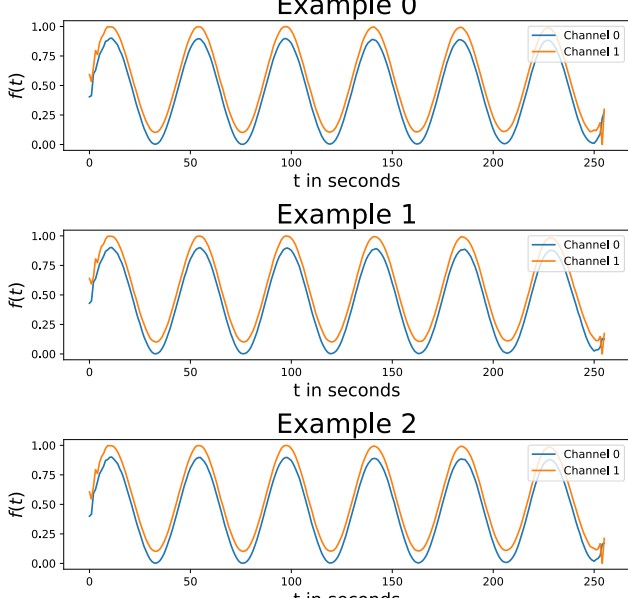

(a) Output of a GAN model with an LSTM generator and TCN discriminator which resulted in a mode collapse. The produced time series are sine waves, but all generated examples are very similar.

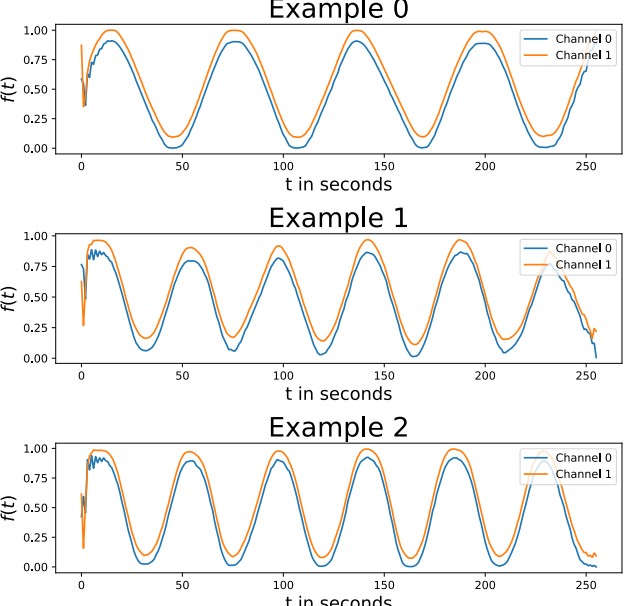

(b) Output of a GAN model with a TCN generator and TCN discriminator which resulted in no mode collapse. The produced time series are sine waves and the generated time series contain different frequencies.

Figure 24: A mode collapse during training in the sine data set, notice how the upper sine wave experience the same periodicity, while the lower example samples differ in their frequency.

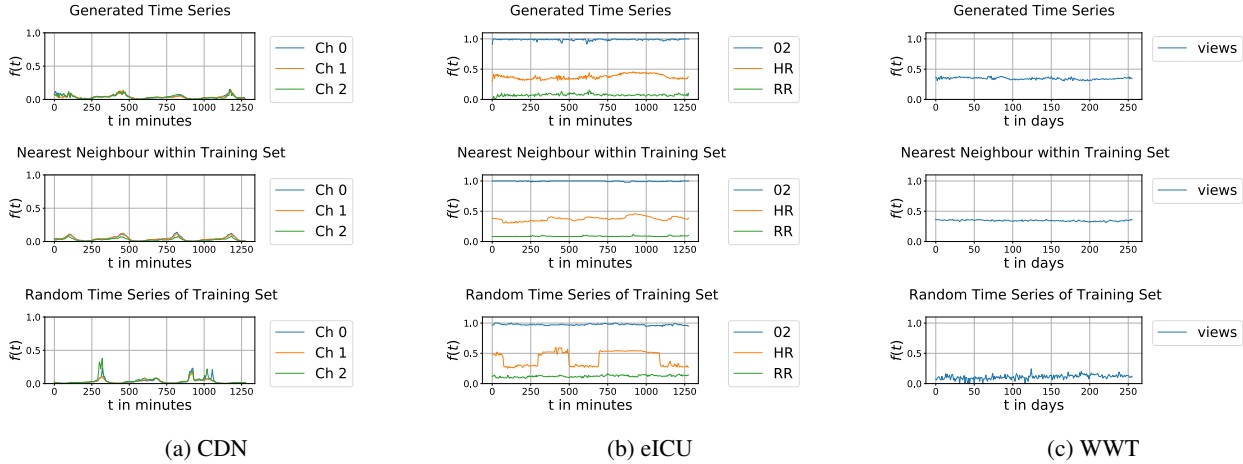

(a) CDN         (b) eICU         (c) WWT

Figure 25: Visualization of generated examples by GANs with a TCN generator and TCN discriminator trained on the data sets. For each generated time series, its nearest neighbour and random example of the training are displayed trained data set. For the distance calculation, the ED is used.

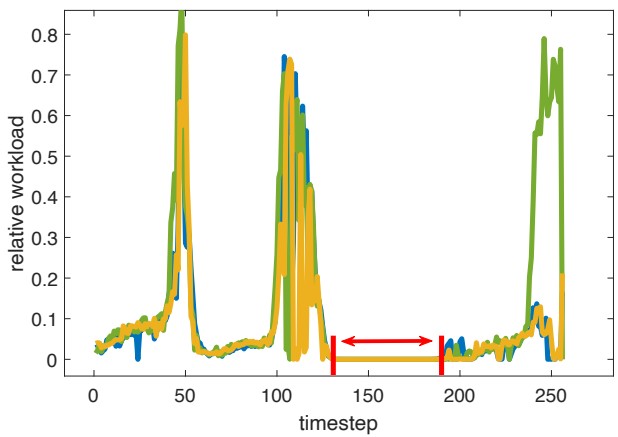

Figure 26: Collective anomalies synthesis on the CDN data set using the TCN GAN setup. The anomalies, corresponding to an outage, are inbetween the red lines.

turing the daily views of another article is different with regard to the average daily views.

**Overview** Overall, the TCN models were able to generate time series with high fidelity for all data sets. In addition, when not experiencing a mode collapse, the generator did not copy the time series of the training set, but rather learned the underlying distribution. Our qualitative evaluation is able to underline findings of the quantitative evaluation in regard to the overall rank scores, and the ability to recognize mode collapse issues.

Noticeably, the performance of a full TFT GAN model varied a lot with regard to the data sets. We analyzed the Pearson correlation coefficient and the FFT comparison to investigate this issues. We found, that the average squared difference for the Pearson correlation and the FFT metric is high for the sine and CDN data set for the TFT model, suggesting a high spatial correlation and seasonalities as the issue in these data sets.

## 7.4 Resource Consumption

We investigate the resource consumption of the different GAN architectures. We assume, that TCN-based models consume less memory and the parallelization of the computation results in faster training times. To verify these assumptions in a GAN context, we compare the required resources to train the different GAN models.

For this, we consider configurations with the same alternate training setup, as configurations with an increased amount of generator training steps will scale linearly and result in a higher run time per epoch. Hence, the average memory consumption and training time of all configurations with one discriminator step and one generator step are averaged for each architecture.

A full TCN setup requires roughly half the amount of resources compared to a full LSTM setup. A full TFT setup requires roughly ten times more resources than TCN and five times more resources than LSTM. Hereby, the sheer compute power and time required for a TCN setup, shall be taken into account when considering an architecture, as it does, as of now, offer the return on investment in regards to data fidelity versus compute time and allocated resources. The results are listed in Table 6.

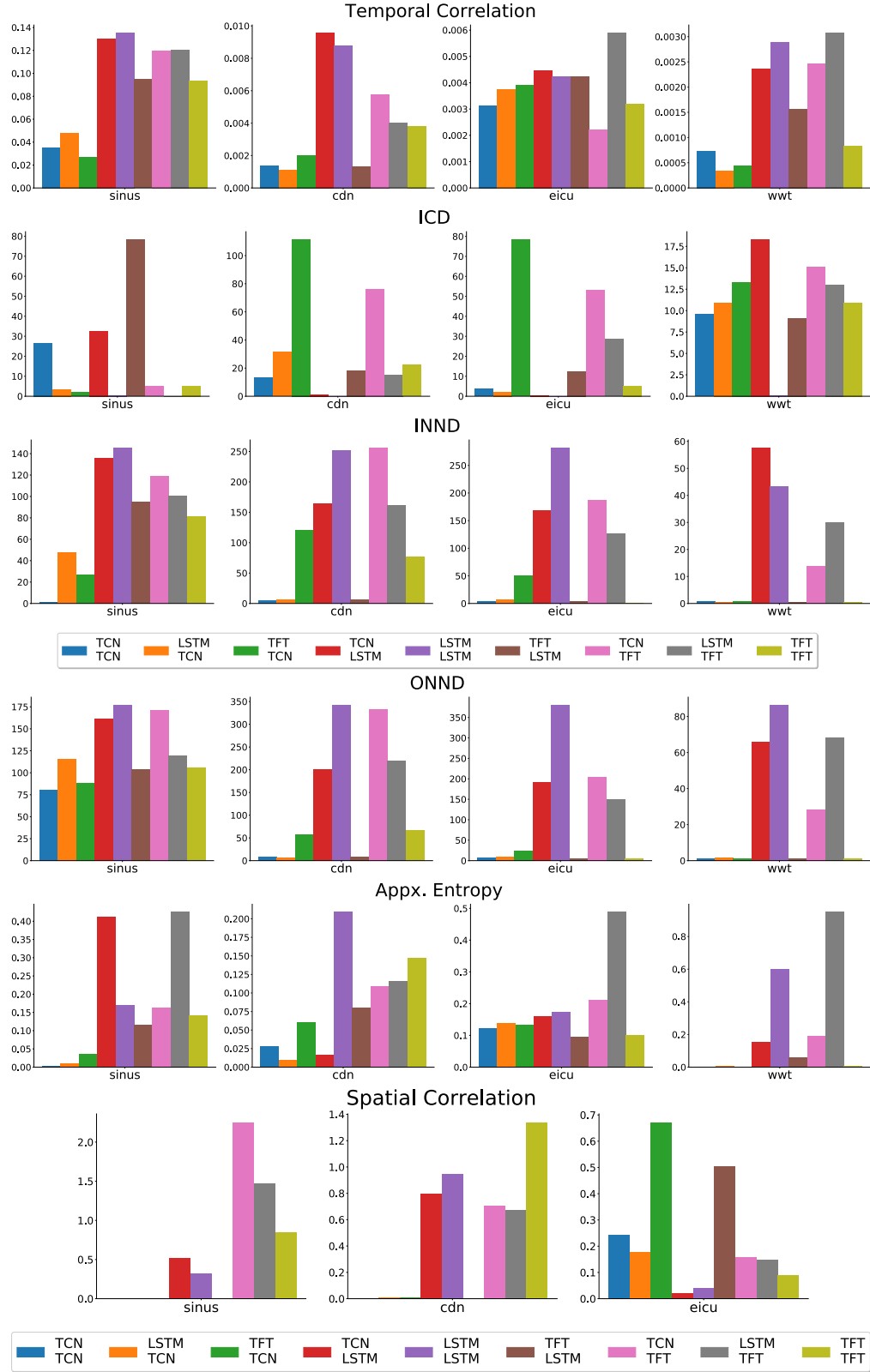

Figure 27: Results overview of the evaluation metrics computed for all architectures on the four data sets. A lower value indicates a better performance for all metrics except acINND.

Table 6: Average run time per epoch in seconds and average GPU memory consumption in GB while training GAN models with different architectures. Architectures with the least resource consumption are marked in **bold**. Values are averaged over four/eight (Sine/CDN data set) configurations with the same alternate training setup.

| G | D | Sine Time per epoch | Sine Memory consumption | CDN Time per epoch | CDN Memory consumption |
|---|---|---|---|---|---|
| TCN | TCN | **6.36s** | **2.39** | **5.5s** | **2.4** |
| TCN | LSTM | 9.32$s$ | 2.69 | 8.38$s$ | 2.71 |
| TCN | TFT | 35.42$s$ | 9.40 | 33.07$s$ | 9.22 |
| LSTM | TCN | 14.15$s$ | 4.69 | 8.59$s$ | 5.01 |
| LSTM | LSTM | 11.35$s$ | 4.55 | 10.61$s$ | 4.74 |
| LSTM | TFT | 37.42$s$ | 11.33 | 36.24$s$ | 11.38 |
| TFT | TCN | 23.11$s$ | 17.17 | 18.70$s$ | 15.29 |
| TFT | LSTM | 25.72$s$ | 17.25 | 21.92$s$ | 15.36 |
| TFT | TFT | 51.85$s$ | 23.25 | 47.37$s$ | 21.27 |

## 8 Discussion

In the following, we focus on providing context for the empirical results of Section 7, discussing the results in regards to the related work. We further evaluate the usability and threats to validity of our approach. Lastly, we debate the generalizability of our results.

### 8.1 Evaluation Metrics

We have focused on comparing the performance of different GAN models by utilizing metrics from several domains. Most state-of-the-art approaches evaluate the performance of their GAN models with a limited set of measurements. By employing measurements from the field of time series analysis and time series similarity, this work was able to evaluate the performance of GAN models with regard to temporal correlations, spatial correlations, mode collapse issues and nearest neighbours of an unknown distributions.

E.g, Esteban, Hyland, and Rätsch [10] did not consider any entropy metrics to measure irregularities and noise in the generated time series. Additionally, the authors did not utilize an auto-correlation or DFT to consider the temporal correlations of the generated time series. Lin et al. [27] did not measure irregularities and noise in the generated time series. Leznik et al. [24] were the only authors to consider entropy measurements for the evaluation task. All the authors did not utilize a metric to account for mode collapse issues while evaluating their GAN models. In contrast, our work was able to detect mode collapse issue with the ICD measurement.

### 8.2 Architecture Comparison

Considering the results of the four synthesised data sets, two clusters appear. The first cluster consists of the artificial sine data set and the CDN data set. In this cluster GAN models with a TCN discriminator achieved the best data synthesis, while LSTM and TFT based discriminators struggled to synthesise time series with a high fidelity and flexibility.

The second cluster consists of the WWT and eICU data set. In contrast to the first cluster, a full TFT setup resulted in generated time series with higher fidelity and flexibility in this cluster. For the eICU data set the full TFT setup performed well in all evaluation metrics.

In order to understand, why the full TFT setup struggled to synthesise realistic examples for the sine and CDN data set, we analyzed our measurement scores. Noticeably, the TFT setup was not able to learn the temporal and spatial correlations as suggested by the FFT and correlation metrics. In contrast, GAN models with a TCN discriminator were able to capture those patterns. For the WWT and eICU data set, the full TFT setup was able to achieve scores similar or even better than the TCN-discriminator-based models (Table 5). This is visualized in Figure 27. This suggests, that the TFT model was not able to capture the spatial correlation and seasonalities of the time series in the first cluster. Considering the analysis done in section 5, the sine and CDN data sets contain periodic time series with high spatial correlations. In contrast, the WWT and eICU data set contain less seasonalities and a lower spatial correlations (none for the univariate WWT data set). Based on these results, we argue, that the TFT model was not able to synthesise periodic time series with a high spatial correlation. However, for data sets, that do not fulfill these characteristics, the TFT GAN model was able to produce time series with high fidelity and variation.

Over all data sets a full TCN setup was the most stable architecture. The GAN models trained with this architecture were the only models to not suffer from mode collapse.

For this work we followed the approach by Isola et al. [17] to not deactivate the dropout layers while generating time series for evaluation purposes. We applied this for all architectures, expect the LSTM generator, which does not contain a dropout layer. Here, we noticed, that the added noise by this approach resulted in higher fidelity in the generated time series. When deactivating the dropout layers the generator produced time series with low variance. While Isola et al. [17] synthesised images, we argue that it is beneficial for time series GANs to follow this approach.

Generally, the LSTM discriminator did not perform well due to vanishing gradient problems. These problems occurred especially when using a sigmoid output layer in the generators in order to achieve a time series in range of $[0; 1]$. We have tried multiple modifications to the LSTM architecture, but none could resolve this problem. For example, we tested bidirectional as well as directional LSTM layers, followed up by different approaches to achieve the desired classification task.

We chose a fixed sequence length of 256 for all data sets. When decreasing the sequence length it has to be expected that the performance of the LSTM architectures and TFT architectures, which contain an LSTM block, would improve.

The results we provide can be seen as an extension of the prior work from Bai, Kolter, and Koltun [3]. While Bai, Kolter, and Koltun [3] provided empirical results for TCN performance in time series tasks, this work extends these results for the task of data distribution estimation of time series via GANs and it also considers transformer-based architectures.

The results indicate, that the performance of these GAN models depend on the discriminator architecture. Overall, we suggest to use a TCN generator and TCN discriminator for unknown time series synthesis. When considering resource consumption, the TCN models could be trained twice as fast with half the amount of memory required compared to LSTM based models. Our suggested transformer model were inefficient compared to the other architectures. Further, the TCN models provided the most stable time series synthesis for all data sets (see Figure 27). This includes the fidelity and variation of the generated time series.

## 8.3 Threats to Validity

**Squared Difference Calculation**    A GAN model implicitly estimates an unknown distribution, however, only generated examples can be utilized to evaluate the performance. The ground-truth data distribution, which should be estimated, is represented by a set of training examples. The size of this set is limited by definition. Therefore, the two data distributions can only be compared by computing measurements on the samples. For this task, we generated 10 random examples at each epoch while training the GAN networks. The measurements of time series analysis and time series similarity were computed and averaged over these 10 examples. For the training data set, 500 random examples were taken to compute the same metrics for the training data set. After averaging the metrics over the 500 examples the squared difference was computed.

The accuracy of comparing two distributions based on metrics computed on samples is correlated to the number of samples used for the comparison task. Increasing the number of samples would remove noise from our results.

The 500 examples taken to compute the metrics for the training data set were also used to train the GAN. For other ML applications, it is common to use a separate test split to evaluate the performance. In the context of a GAN model, where the task is to estimate the unknown data distribution, we argue, that evaluating the performance based on training examples does not introduce a bias, as the number of training sample used for this task is small compared to the overall data set size. With this approach, it is unlikely, that the generator part of the GAN models overfits these random samples. We account for possible overfitting to the training data with our quantitative and qualitative analysis.

The assumption for our squared difference approach to work is, that the average value of a metric extracted from samples can represent the data set well. For the data sets,

which were used in the comparison task, this assumption is fulfilled.

For data sets, which do not fulfill this assumption, encapsulating the characteristics of a data set by an average value loses information. We can counteract this information loss by interpreting the metrics extracted per sample as a distribution. Those values sampled from an unknown distribution function could then be compared to the samples from the other distribution. The Kullback-Leibler-Divergence is an example to measure the similarity between two distributions [23].

**Fast Fourier Transformation Analysis**    Our FFT evaluation metric only compares the peaks in the frequency spectrum. This hinders the framework to detect if the generated time series contains frequency components of similar frequency bands. To discuss if this could bias the results, the characteristics of the two data sets with high seasonalities are used.

For the CDN data set, all extracted time series contain the same seasonalities and the generated time series contain the peaks at the same frequency bands. This can be seen in Figure 25a, where a generated time series contains the same daily cycles as the training time series. Therefore, it was enough to only compare the peaks in the frequency spectrum. A well defined sine wave only contains a single frequency component. Therefore, it is enough to evaluate the performance in the frequency spectrum by peak values.

**Privacy Measurements**    We did not consider privacy measurements for the empirical results The performance of the architectures was compared with regard to fidelity and variability of the generated time series. The INND and ONND metrics could be leveraged in future work to analyze if the generated time series leak sensitive information.

Additionally, the GAN architecture could be modified to use differential privacy [9] (DP) for privacy-preserving. For this task, the gradient descent utilizes clipped and noisy gradients to not over-fit certain examples. However, Lin et al. [27] argue, that current DP mechanisms require improvements to be utilized in time series GANs.

**Transformer Architecture**    Initial results indicate, that transformer models as proposed by Vaswani et al. [35] are not suitable for time series data. Therefore, we modified the TFT architecture as proposed by Lim et al. [26] for the GAN architecture. The TFT architecture contains an LSTM block, and it is unclear, if this should be characterised as a transformer model with an LSTM component, or a complex RNN model. Lim et al. [26] argue that the LSTM component processes local information, while the multi-head attention mechanisms is utilized to integrate this information by the self-attention-mechanism. We argue, that this TFT model should be seen as a modification of RNNs, which utilizes the self attention

mechanism to counteract the issues of vanishing gradients for long time series tasks. This might also explain, why the TFT discriminator did not suffer from the same vanishing gradient problems as the LSTM discriminator. The training time of the TFT transformer may be attributed our training approach. Li et al. [25] argue that training wider and deeper models, at least for NLP use cases, converges faster and lead to more robust models.

**Parameter Search** To limit the scope of the parameter search, we fixed some parameters in advance. For all GAN models the same GAN training parameters were used. This includes the optimizer, learning rate, latent space, loss function, batch size and number of epochs.

For the TCNs the number of layers and kernel size was fixed to achieve a receptive field size, which covers the whole input sequence. It is not clear how the performance of the TCN models varies with an even higher receptive field size. For the LSTM models and TFT models a fixed number of hidden neurons in the LSTM block was used. Further parameters such as the embedding dimension, dropout and attention heads were not varied for the TFT models.

We chose these configurations after analyzing preliminary results based on visual evaluation on the sine data set, assuming the selected parameters are transferable to the other data sets. According to Lucic et al. [28] this might not always be the case.

**Data Sets** We provide results for four data sets of varying characteristics. It has become apparent, that the artificial sine data set is especially prone to mode collapse issues. This may be caused by the normalization of each training example. After applying the pre-processing the sine data set has a low variance. This contradicts the proposal of Lin et al. [27] to normalize data sets in order to decrease the variance and counteract mode collapse issues. We assume, that their approach only helps data sets with extreme outliers. As many real data sets are also prone to mode collapse issues, this setup provided relevant empirical results as to which architectures are more stable.

## 9 Conclusion

This work proposed multiple metrics to counteract the common problem of evaluating the performance of a data distribution estimation task for time series to account for different patterns and characteristics of the data.

These metrics were then used to compare the performance of common neural network architectures used for time series generation tasks. We evaluated the suitability of the metric by qualitative and quantitative analysis.

We provided empirical results for the comparison of the architectures in a GAN context for time series tasks for multiple data sets, which has not been done so far.

The empirical results we provide indicate that a TCN architecture is beneficial for GAN models due to efficiency, performance and training stability. As the TCN GANs were successful for all data sets with varying characteristics, it can be expected that the empirical results are generalizable to other data sets. The LSTM discriminator used suffered immensely from vanishing gradient problems. This problem was not present in the LSTM generator part. The empirical results indicate that the TFT model struggled to synthesise time series with high spatial-correlation and seasonalities, and is hence only suitable to synthesise time series with low temporal and spatial correlations. For these data sets, a full TFT GAN resulted in the best time series synthesis. However, the resource consumption of the TFT architecture is immense compared to TCN models.

To verify the empirical results with regard to different GAN and ANN parameters, future work could include further parameters in the parameter search to investigate the influence of different optimizer, loss functions and architecture parameters on varying data sets and architectures. While we chose a grid search to cover all possible configurations, future work could consider applying a random search instead. A bigger number of parameters could be investigated by utilizing a random search.

Overall, our contribution includes a rigorous extensive evaluation of multiple GAN architectures for the purpose of time series synthesis We propose an approach to combine time series specific measurements, such as similarity, spatial and temporal correlation and GAN specific measurements, such as the possibility to detect mode collapse during the training process. We provide a clear structured approach for future research to apply to a given data set before settling on the target GAN architecture.

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
