# OpenReview forum: "SoK: The Great GAN Bake Off, An Extensive Systematic Evaluation of Generative Adversarial Network Architectures for Time Series Synthesis"
_JSYS/2022/Feb_Papers — Submitted to JSYS Feb 22_

### Official Review · Reviewer_5Yiu · 2022-03-03
**Well-written paper on comparing different GAN architectures for generating time-series data but it needs more experiments to confirm the conclusion**

**Decision:**

Weak accept: good paper with flaws that can be fixed in three months

**Review:**

The paper provides an overview of using different GAN architectures for generating time-series data and evaluating them. To this end, the authors presented three families of GAN architectures, based on RNN, CNN, and transformer, and evaluated them using four synthetic datasets with different characteristics. The paper is well-written, nicely reviews the state-of-the-art, and the experiments give some insights into which GAN architecture best fits the provided datasets. The presented datasets cover various characteristics of time-series datasets, such as temporal/spatial correlations and multi-variety, and the empirical results show that the CNN GAN outperforms the other architectures. However, considering different properties of time-series datasets, such as spatial correlation and periodic features, the authors need to conduct the experiments on multiple datasets with similar properties to conclude correctly. For example, the authors currently consider only one high spatial correlation, one low, and one perfect. Thus, the results are not convincing enough as they are just produced based on one dataset, and more importantly, all the conclusions are based on practical observation, not theoretical. Thus, I strongly recommend including more datasets in the experiments. Moreover, it would also be beneficial to study how any of the presented architectures generate rare events in time-series data.

Some minor comments:
page 1, right col.: the predominant architecture used for data synthesis The -> the predominant architecture used for data synthesis. The
page 2, left col.: We evaluate 9 different GAN architectures -> We evaluate nine different GAN architectures
page 10: decomposing time series needs more explanation by defining each of the "T: trend", "S: seasonal", and "E: residual" parts.
page 12, left col.: model. I.e, how -> model. i.e., how
page 12, right col.: The analysis shows, that the CDN -> The analysis shows that the CDN
page 12, right col.: This means, that the amount of -> This means that the amount of

**Expertise:**

Follow the literature closely, last published 5+ years ago

**Useful:**

yes

---

### Official Review · Reviewer_5Zmy · 2022-03-10
**Timely problem with unsatisfactory analysis**

**Decision:**

Weak reject: interesting papers with flaws, not sure if they can be fixed in three months

**Review:**

Pros:
- timely problem on time-series data generation
- considerable efforts with thorough architecture comparison

Cons:
- mostly presenting the results, while analysis missing
- needs better organization and writing of the paper
- the relationship to the system research also needs better articulation

Thank you for submitting your paper to JSys!
This work does provide a thorough analysis over the structure of the GAN in generating faithful and flexible time-series data.
I really like the problem, which is very important since races in systems are usually time-series, and system research always confront the problem of lacking of real-world.
Time-series data plays a significant role in the system research, in scheduling, load balancing, rate control, congestion control, etc.
Therefore, faithfully and flexibly generating time-series data should definitely help the researchers here.
I can also see considerable efforts of the authors' from the experiment parts.

However, I've got a little disappointed after reading through 20 pages and finally to the results section.
The authors straightforwardly presented the results, without too much analysis on *how would it behave like that*.
For example, TCN performs the best on the CDN and WWT dataset. Why is that? Are there any underlying findings about the behaviors? Is it because the periodicity of the dataset. If so, how common is the periodicity for system datasets in general?
For example, in the network traces in congestion control (https://arxiv.org/abs/1802.08730), one can hardly see periodicity unless explicitly adding periodic signals.
I strongly suggest the authors to dig deeper into the underlying reasons rather than directly present the results.

Besides, another major concern on this work is the scope of the paper, i.e. how it could help the researchers in the system community to better understand their design choices and underlying insights.
The authors have already done some awesome analysis over the system use cases, such as the CDN and WWT ones, which, however, are missing in the abstract and intro part.
In this way, I was feeling like reading an JMLR paper until the section 5.
I strongly suggest the authors to better organize the paper in a way that is more related to system research.
The authors did demonstrate the analysis and performance over datasets in the system area (namely, the CDN one and WWT one), yet I do not see significant relationship in the analysis, as in my review above.

As a third-party view comparison paper, it would also be better to test with longer time series.
Even time series with 500 time steps might not be long enough in practical settings.
For example, congestion control algorithms in networking are usually tested with at least hundreds of seconds (tens of thousands of round-trip time, or time steps for algorithms).
Cluster scheduling problems also need much longer traces to evaluate.
It would be very interesting how these algorithms behave in a longer scale.

For the settings of the comparison, some choices seem to be casual.
For example, it would be better to compare multiple neural net microarchitecture settings (connections, etc.) even for one neural net architecture.
Especially it is unclear why one architecture outperforms another in certain datasets in this paper, analysing more neural nets might be helpful to dig out the underlying reasons.

I also suggest the authors to better summarize their main findings in the main body, and leave some detailed descriptions (e.g., over the dataset, on the background information of GAN) to the appendix.
The paper, in its current shape, is a little bit distracting by placing unimportant backgroud knowledge here and there.
I'm feeling that the authors devoted too many efforts in introducing the background and related work, which are way too detailed for the contribution of this paper.
For example, the authors introduce [10] and [26] in Section 3.1 for almost one page, which is rarely to see even in a survey paper.
The true findings might be therefore buried.

Nits:

- Typos
-- Sec. 1, para. 2, "time series generation however, is an the lack" --> "time series generation, however, is the lack"

- Some references have been published in conference proceedings or journals. Please update the citation. For example, [5] in ICLR'19, [11] in NeurIPS'14, [12] in NeurIPS'17, [18] in NeurIPS'21, and [26] in IMC'20.


**Expertise:**

Published in this area in the last 5 years

**Useful:**

yes

---

### Official Review · Reviewer_2WmZ · 2022-03-20
**Overall I liked the work. It provides a large amount of background readers can learn from, there is  attention to detail and the results provide useful guidance for anyone selecting between approaches for time series generators. Perhaps the only nit is that it missed neural ODE based approaches and did not provide sufficient exposition on VAE based approaches.**

**Decision:**

Weak accept: good paper with flaws that can be fixed in three months

**Review:**

Some comments and questions:

Introduction: Can you give some examples of where data from generative models has been used to successfully train/improve standard classification or regression models? Another value proposition of generative models is to give insight about the latent variable structure of the data, eg. with reversible normalizing flows.

Introduction: In addition to compute time spent on the experiments it would be helpful to talk about the number of data points in time series you generated and studied.

Sec 2.1: Can you elucidate a bit more about why VAEs have limitations in estimating probability distributions for time series? I should note that they have been successfully applied in end to end tasks eg. anomaly detection eg (one of many) https://arxiv.org/abs/1802.03903

Sec 2.1: "in a mini-max game". Strictly speaking the training process does not "play" the mini-max game. The mini-max game formulation guarantees a nash equilibrium towards which the training is trying to converge. A few sentences here summarizing the approach of Goodfellow et al would be useful for the interested reader.

Sec 2.2: Does considering only equidistant data points limit the applicability of your work in some way? Note that this is not a limitation when using, for instance, neural ODEs eg. see section 5 of the “Neural Ordinary Differential Equations” paper.

Sec 3.1: Can you provide a few sentences summarizing how the Doppelganger architecture overcomes the limitations of RNNs?

Sec 5.1: What did you mean by normalizing the data? Sin(x) and Cos(x) are bounded [-1, 1] so it's not clear what normalizing you need here.

Sec 5.2: Which one of the FFT components led you to conclude an increasing trend? Or was it the long tail of the spectra?

Sec 5.2: A sliding window approach with a slide of 1 leads to significant correlation between the time series samples. Could this be an issue?

Table 1: Please define the channel acronyms (RR etc) to enable the reader to interpret the table better.

Sec 6.1: The TFT was geared to incorporate covariates from features in the input eg. “was it a holiday?” - in your case however you only use the time step as an input “feature”. Can you talk a bit about whether there might be “overkill” in using a TFT in this situation?

Table 4: It would be useful to know the number of parameters for each setting - this would probably let you directly correlate with the results in Table 5.

Applicability: You seem to have rejected the VAE approach to time series generation altogether. However it is actually easy to evaluate the fidelity of the generator by using the full setup and measuring reconstruction loss. It would appear all this work to measure fidelity of generated time series applies only to GAN approaches. Can you include some discussion of this in your background and expand on VAEs?





Nits/typos:

- Abstract: "Comobinations"
- Introduction: "Generating or extending large" ?
- Introduction: "Poised stem"
- Sec 2.1: "Considering the present data distribution" ... I think you meant D(x, \theta_D) ?
- Sec 3.1: “Dilution factor” -> Dilation factor
- Sec 5.1: “Sinus” -> “Sine”, “CoSinus” -> “Cosine”
- Eq 8,9, 10. For consistency, there should be an additional subscript to show that f is being restricted to channel d, or perhaps f should take the channel as a parameter in Eq 7.
- Sec 6.2.2 Please define “ED” before you use the acronym. I gathered its Euclidean Distance but this is just a guess that is consistent with the rest of the text.


**Expertise:**

Published in this area in the last 5 years

**Useful:**

yes